# Metaxins are core components of mitochondrial transport adaptor complexes

Yinsuo Zhao[1,2], Eli Song [1], Wenjuan Wang[1], Chung-Han Hsieh[3], Xinnan Wang [3], Wei Feng [1,2✉], Xiangming Wang[1✉] & Kang Shen [4✉]

Trafficking of mitochondria into dendrites and axons plays an important role in the physiology and pathophysiology of neurons. Mitochondrial outer membrane protein Miro and adaptor proteins TRAKs/Milton link mitochondria to molecular motors. Here we show that metaxins MTX-1 and MTX-2 contribute to mitochondrial transport into both dendrites and axons of *C. elegans* neurons. MTX1/2 bind to MIRO-1 and kinesin light chain KLC-1, forming a complex to mediate kinesin-1-based movement of mitochondria, in which MTX-1/2 are essential and MIRO-1 plays an accessory role. We find that MTX-2, MIRO-1, and TRAK-1 form another distinct adaptor complex to mediate dynein-based transport. Additionally, we show that failure of mitochondrial trafficking in dendrites causes age-dependent dendrite degeneration. We propose that MTX-2 and MIRO-1 form the adaptor core for both motors, while MTX-1 and TRAK-1 specify each complex for kinesin-1 and dynein, respectively. MTX-1 and MTX-2 are also required for mitochondrial transport in human neurons, indicative of their evolutionarily conserved function.

[1] National Laboratory of Biomacromolecules, CAS Center for Excellence in Biomacromolecules, Institute of Biophysics, Chinese Academy of Sciences, 15 Datun Road, Beijing 100101, China. [2] College of Life Sciences, University of Chinese Academy of Sciences, Beijing 100049, China. [3] Department of Neurosurgery, Stanford University School of Medicine, Stanford, CA, Stanford, CA, USA. [4] Howard Hughes Medical Institute, Department of Biology, Stanford University, Stanford, CA, USA. ✉email: wfeng@ibp.ac.cn; xmwang@ibp.ac.cn; kangshen@stanford.edu

Neuronal mitochondria play important roles in neuronal physiology. As the cellular power plants of neurons, mitochondria provide ATP to sustain the firing of action potentials and continuous synaptic transmission[1,2]. As a cellular calcium reservoir, mitochondria regulate axonal cytosolic Ca$^{2+}$ level[3]. In addition, mitochondria participate in producing many biosynthetic intermediates, as well as cellular stress responses such as autophagy and apoptosis[4,5]. The extreme size of neurons creates challenges for ATP diffusion[6]. Hence, neurons have evolved specialized mechanisms to efficiently transport mitochondria to distal processes where energy is in high demand[2,7–9]. Impairment of mitochondrial trafficking in neurons results in neuronal development and maintenance defects, and human neurodegenerative diseases[10–12].

Mitochondrial transport in axons and dendrites is mediated primarily by the microtubule-dependent motor systems, including both kinesin and dynein[13]. Kinesin drives mitochondria towards the plus-ends of microtubules, while dynein moves them in the opposite direction. Among the kinesin superfamily, KIF5/kinesin-1 and KIF1B/kinesin-3 are the primary kinesins that mediate mitochondrial transport in neurons[14]. Mitochondria recruit molecular motors through membrane adaptor proteins, which not only mediate the movement of mitochondria, but also dictate how neuronal activity and extrinsic signals regulate the abundance of mitochondria in axons[2,15]. Previous studies have shown that the adaptor complex contains two components: the mitochondrial outer membrane atypical Rho GTPase Miro[16], and the adaptor protein Milton in *Drosophila* (TRAKs or OIP106 and GRIF-1 in mammals)[17,18]. Importantly, the Miro-TRAK complex plays critical roles in linking mitochondria to both kinesin and dynein motors[2,9]. In mammalian cells, two TRAK proteins function differently to steer polarized transport of mitochondria: TRAK1, which is mainly distributed in axons, binds to both kinesin-1 and dynein, and is required for axonal localization of mitochondria, whereas TRAK2, which is mainly localized in dendrites, primarily interacts with dynein and is responsible for the dendritic distribution of mitochondria[19].

Miro is localized to the outer mitochondrial membrane through the C-terminal transmembrane domain and contains two GTPase domains that are distantly related to the small GTPases of the Ras superfamily[2]. In *Drosophila*, *dmiro* mutants showed dramatic defects in mitochondrial trafficking to axons and dendrites[16]. Further studies have shown that Miro anchors mitochondria to microtubule motors through binding to Milton/TRAK[20,21]. Besides, Miro might also bind to KIF5/kinesin-1 directly[22], and recruits the mitochondrial myosin Myo19 to link mitochondria to the actin cytoskeleton[23]. Miro contains a pair of Ca$^{2+}$-binding EF-hands, which sense neuronal activity or oxidative stresses to arrest mitochondrial trafficking[24]. Miro is also the target of the Parkinson's disease-associated gene PINK1, which phosphorylates Miro and triggers its degradation through the ubiquitin ligase Parkin[25].

Is Miro the only outer membrane protein that links mitochondria to motor proteins? Several pieces of evidence suggest that this is unlikely. First, loss of *Drosophila* Miro (dMiro) cannot fully block mitochondrial movement: small amounts of mitochondria are still localized in neurites[16,26]. In rodent hippocampal neurons, the distribution of mitochondria along the axon is not dependent on Miro1, although fast axonal mitochondrial trafficking is severely affected in Miro1 knockout neurons[11]. The TRAK1 and TRAK2 adaptor proteins are still recruited to the outer mitochondrial membrane to drive mitochondrial trafficking in Miro1/2 double-knockout cells[23]. These data indicate that the previously unknown adaptors, probably independent of Miro, exist on the mitochondrial outer membrane to recruit TRAK or other proteins to mediate mitochondrial trafficking.

To identify potential adaptor proteins, here we establish an in vivo system to investigate mitochondrial distribution and trafficking in axons and dendrites using the PVD sensory neuron of *C. elegans*. The PVD neuron elaborates a single axon, one anterior and one posterior dendrite with highly ordered branches[27]. Our previous work has shown that microtubule organization is uniform along the anterior–posterior (A-P) axis of PVD[28]. Microtubules in the anterior primary dendrite are predominantly oriented with their minus-ends pointing towards the distal dendrite. In contrast, the posterior primary dendrite and the axon mainly contain plus-end out microtubules. Thus, dynein transports cargo towards the tip of the anterior primary dendrite, whereas kinesins transport cargo towards the distal posterior dendrite and axon. Using cell type-specific promoters, we have visualized mitochondria in PVD neurites with single-cell resolution.

During a forward genetic screen for mutants with abnormal PVD dendritic mitochondrial distribution, we isolated the *mtx-2* mutant, which abolished dendritic mitochondria. We further show that the known binding partner of MTX-2, MTX-1, is also required for mitochondrial trafficking. The metaxin complex consisting of metaxin1 (called SAM37, MAS37, or TOM37 in yeast) and metaxin2 (called SAM35, TOM38, or TOB38 in yeast), functions with SAM50 to form the mitochondrial sorting and assembly machinery to fold and insert β-barrel proteins onto the mitochondrial membrane[29]. Our data show that the metaxin complex directly binds to the kinesin motor and MIRO-1 to control mitochondrial trafficking. Our genetic and biochemical data suggest that there are two adaptor complexes: an MTX-2/MIRO-1/MTX-1/KLC-1 complex responsible for kinesin-mediated mitochondrial movement and an MTX-2/MIRO-1/TRAK-1 complex responsible for dynein-mediated mitochondrial trafficking. Finally, MTX-1 and MTX-2 homologs mediate mitochondrial trafficking in human iPS-cell derived neurons, revealing that their functions in mitochondrial trafficking are evolutionarily conserved.

## Results

**MTX-2 and MIRO-1 are essential for mitochondrial localization in PVD axon and dendrites.** To study trafficking mechanisms of mitochondria in axons and dendrites, we visualized mitochondria by tagging the mitochondrial outer membrane protein TOMM-20 with GFP in the *C. elegans* sensory neuron PVD (PVD > TOMM-20 (1-54AA)::GFP), together with PVD > mCherry to label the PVD neurites (Fig. 1a). In addition to the intense GFP signal in the PVD soma, round, oval, and occasionally rod-shaped mitochondria were detected as individual punctum throughout the highly elaborate anterior and posterior dendrites (Fig. 1a, b) as well as in the axon (Fig. 1a). Additional mitochondria markers showed a similar pattern as the TOMM-20 marker (Supplementary Fig. 1a, b).

A large body of evidence has shown that the conserved mitochondrial protein Miro is required for mitochondrial trafficking. We used Cas9 to generate a null mutation in *miro-1*, the *C. elegans* orthologue of Miro[30]. In the *miro-1* mutant, PVD dendrites were devoid of mitochondria (Fig. 1b–d), confirming the requirement of the Miro homologue for proper mitochondria trafficking in *C. elegans*. To find other molecules important for mitochondrial trafficking, we carried out an unbiased forward genetic screen using PVD dendritic mitochondria as a system. From a visual screen covering 4000 haploid genomes, the mutation with the largest defect in mitochondrial distribution, *wy50256*, showed nearly complete absence of mitochondria from both anterior and posterior dendrites, a phenotype that is indistinguishable from that of *miro-1* mutants.

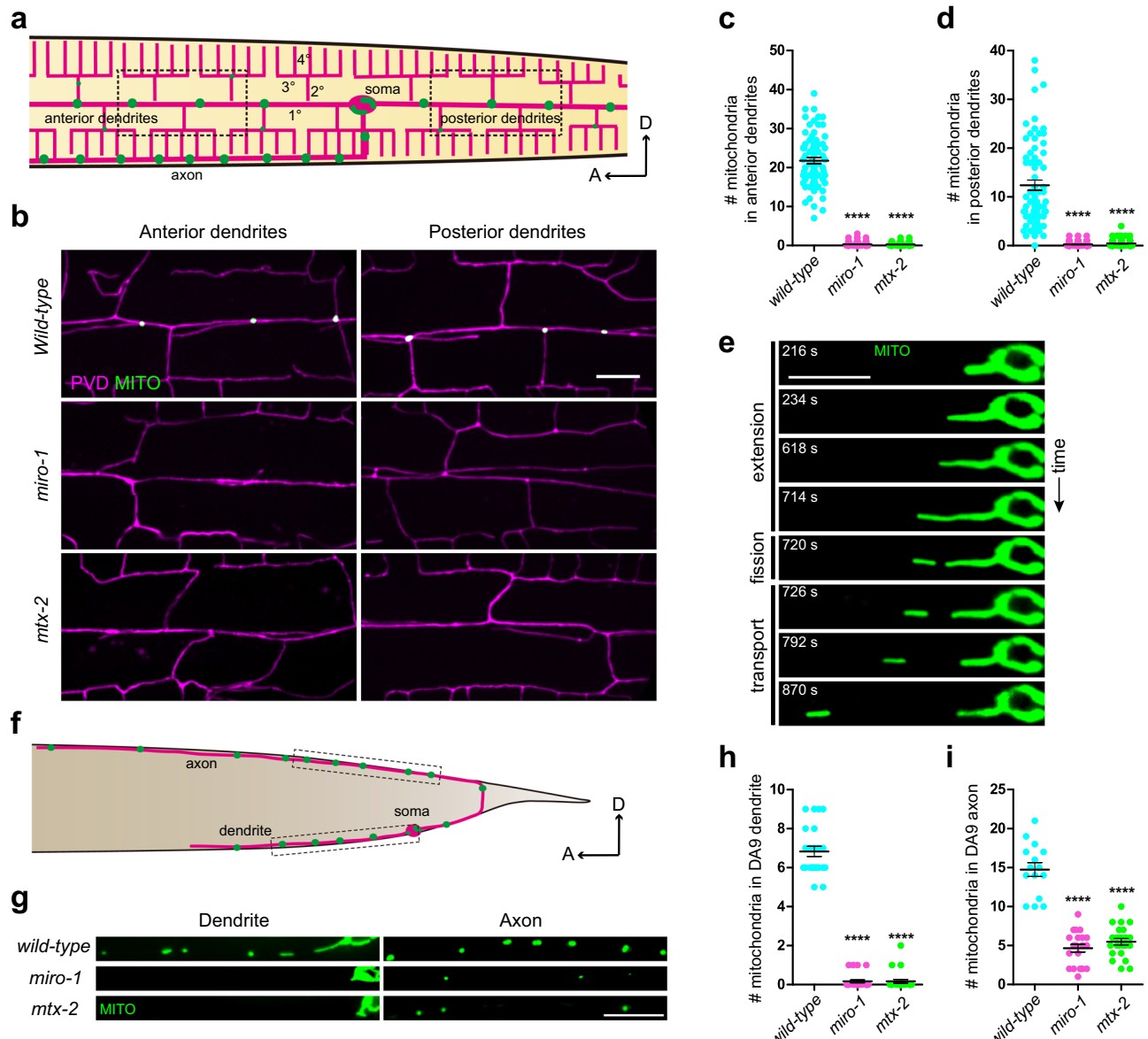

**Fig. 1 MTX-2 and MIRO-1 are essential for mitochondrial localization in axons and dendrites. a** A schematic diagram showing mitochondrial distribution in the axon and dendrites of PVD. Magenta marks the PVD morphology showing a branched dendritic arbor containing highly ordered primary (1°), secondary (2°), tertiary (3°), and quaternary (4°) branches. Green dots along the neurites represent mitochondria. Anterior is to the left and dorsal is up. **b** Representative confocal images showing dendritic morphology and mitochondrial distribution of PVD neuron in wild type, *miro-1*, and *mtx-2* mutants. Magenta: PVD > mCherry. Green: PVD > TOMM-20(1-54AA)::GFP. Scale bar: 10 μm. **c, d** Quantification of the number of mitochondria in anterior (**c**) and posterior (**d**) dendrites of PVD. Data are shown as mean ± SEM. One-way ANOVA with Tukey's multiple comparisons test (95% CI). ****$p < 0.0001$. (exact $p$ values and sample size: wild-type vs. *miro-1*, $p < 0.0001$; wild-type vs. *mtx-2*, $p < 0.0001$. $n = 72$ animals for each genotype). **e** Representative time-lapse images showing "extension", "fission", and "transport" behaviors of mitochondria in the anterior dendrite of PVD. Scale bar: 10 μm. **f** A schematic diagram showing mitochondrial distribution in axon and dendrite of DA9. Magenta: DA9 morphology. Green: mitochondria. Anterior is to the left and dorsal is up. **g** Representative confocal images showing the mitochondrial distribution of DA9 neuron in wild type, *miro-1*, and *mtx-2* mutants. Green: DA9 > TOMM-20(1-54AA)::GFP. Scale bar: 10 μm. **h, i** Quantification of mitochondria number in dendrite and axon of DA9 in wild type, *miro-1*, and *mtx-2* mutants. Data are shown as mean ± SEM. One-way ANOVA with Tukey's multiple comparisons test (95% CI). ****$p < 0.0001$. (exact $p$ values and sample size: **h** wild-type vs. *miro-1*, $p < 0.0001$; wild-type vs. *mtx-2*, $p < 0.0001$. wild-type, $n = 23$; *miro-1*, $n = 24$; *mtx-2*, $n = 24$ animals. **i** wild-type vs. *miro-1*, $p < 0.0001$; wild-type vs. *mtx-2*, $p < 0.0001$. wild-type, $n = 15$; *miro-1*, $n = 20$; *mtx-2*, $n = 23$ animals). Source data are provided as a Source Data file.

With genetic mapping and sequencing, we found that *wy50256* caused a premature stop codon at Q9 of *mtx-2*, which encodes one component of the mitochondrial outer membrane protein complex metaxin, is responsible for this trafficking phenotype (Fig. 1b–d). This phenotype might be due to the lack of mitochondrial trafficking into the neurites or defects in the anchoring or maintenance of mitochondria in neurites. To

distinguish between these possibilities, we performed time-lapse imaging experiments to record the dynamics of mitochondrial trafficking. In wild-type worms, the somatic mitochondrial network gave rise to elongated tubules. As the tubule lengthens ("extension"), a distal tubular segment broke off through a distinct fission event ("fission"). The rod-shaped mitochondrion then migrated away progressively ("transport") (Fig. 1e,

Supplementary Fig. 1c, d, and Supplementary Movie 1). Within PVD dendrites, mitochondria were transported both away from the cell body (anterograde) and towards the cell body (retrograde) while many remained stationary during the course of our imaging experiments (Supplementary Fig. 1e–h). Compared with wild type, both *miro-1* and *mtx-2* mutants showed diminished "extension" in both anterior and posterior dendrites (Supplementary Fig. 1i–l and Supplementary Movie 2). In wild-type controls, about a third of the "extension" events lead to "fission" and subsequent transport. In *miro-1* or *mtx-2* mutants, we did not observe a single "fission" and "transport" events during a total of 14 and 17.5 h of time-lapse recordings, respectively (Supplementary Fig. 1i–l). These data suggest that *mtx-2* functions in transporting mitochondria to PVD dendrites.

To test if *mtx-2* is also required for mitochondrial trafficking in other neurons, we examined mitochondrial localization in the motor neuron DA9. Both *miro-1* and *mtx-2* mutants showed severely reduced mitochondria in both the axon and dendrite of DA9 (Fig. 1f–i), indicating *mtx-2* and *miro-1* are also required for mitochondrial localization into DA9 axons and dendrites. On the other hand, the distribution of early endosome (GFP::RAB-5) and late endosome (GFP::RAB-7) in PVD neurites is indistinguishable in wild type, *miro-1*, and *mtx-2* mutants, further confirming that *mtx-2*'s effect on mitochondrial localization is specific (Supplementary Fig. 1m–p).

The metaxin/SAM (sorting and assembly machinery) complex is thought to be localized on the mitochondrial outer membrane and helps to insert β-barrel proteins into the mitochondrial membrane[29]. We generated PVD-specific expression of a single-copy transgene of GFP fused to MTX-2 to monitor its subcellular localization. We detected GFP signal in the cell body and fluorescent puncta along PVD neurites. Co-expression of TOMM-20::mCherry demonstrates that GFP::MTX-2 signal is co-localized with TOMM-20 (Fig. 2a, b), further confirming that MTX-2 in the worm is localized to mitochondria of neurons.

**MTX-2 can substitute for part of MIRO-1's function when overexpressed**. The similarity between the *mtx-2* and *miro-1* mutant phenotypes argues that MTX-2 might function together with MIRO-1. To understand the relationship between *mtx-2* and *miro-1*, we constructed double mutants between *mtx-2* and *miro-1*. In PVD, mitochondria were completely absent from both anterior and posterior dendrites in the double mutants (Fig. 2c). The double mutant phenotype was not statistically different from those of single mutants because the single mutant phenotypes were already severe (Fig. 2d, e). In DA9 dendrites, the mitochondria were absent in the double mutant as well as in both single mutants (Supplementary Fig. 2a, b). These results suggest that both *miro-1* and *mtx-2* are essential for mitochondrial transport into dendrites. Interestingly, *miro-1* and *mtx-2* single mutants showed a reduced number of mitochondria in the DA9 axon, with about 30% of mitochondria remaining in the axons. In the double mutants, mitochondria were completely absent from DA9 axons (Supplementary Fig. 2a, c), suggesting that MTX-2 can at least carry out some MIRO-1 independent functions.

To further test the genetic interaction between these two genes, we overexpressed each gene in either *miro-1* or *mtx-2* mutants. Expression of *miro-1* or *mtx-2* genomic DNA constructs in PVD in the corresponding *miro-1* or *mtx-2* mutant fully rescued the mitochondria phenotype (Fig. 2c, f, g), indicating that both genes function cell autonomously to transport mitochondria. While overexpression of *miro-1* did not rescue any aspect of the *mtx-2* phenotype (Fig. 2c, f, g), overexpression of *mtx-2* completely rescued mitochondrial localization in the posterior dendrites of *miro-1* mutants, but did not in the anterior dendrites (Fig. 2h–j).

These results argue that although both MTX-2 and MIRO-1 are required for mitochondrial transport under physiological conditions, MTX-2 can substitute for MIRO-1 in certain contexts, but not vice versa.

Despite the fact that MIRO-1 is a tail α-helical anchor protein and not a β-barrel protein, the known role of the metaxin complex makes it plausible that it might help to localize MIRO-1 into the mitochondrial outer membrane. To test this idea directly, we ask if *mtx-2* is required for the mitochondrial localization of MIRO-1 with a strain where GFP was inserted in frame into the endogenous *miro-1* locus using Cas9 (Supplementary Fig. 3a). The wild type GFP::MIRO-1 showed a punctate fluorescence pattern throughout the cytosol of embryos that co-localized with TOMM-20::mCherry (Supplementary Fig. 3a), confirming its localization in mitochondria. The pattern and intensity of GFP::MIRO-1 fluorescence were unaffected in *mtx-2* mutants (Supplementary Fig. 3b, c), indicating that MTX-2 does not function to recruit MIRO-1 to mitochondria. To further support this idea, we performed western blot analysis using the GFP::MIRO-1 knockin strain. Detected by the GFP antibody, MIRO-1 levels were indistinguishable in *mtx-2* mutants and the wild-type control, indicating that the steady-state level of MIRO-1 is not affected in *mtx-2* mutants (Supplementary Fig. 3d, e). This result is consistent with a previous study that *gem1*, the ortholog of *miro-1* in *S. cerevisiae*, is independent of metaxin/SAM50 complex for its mitochondrial localization[31]. We also tested if *miro-1* was required for the mitochondrial localization of MTX-2. In the *miro-1* mutant, GFP::MTX-2 was still localized to puncta along the posterior dendrites of PVD (Fig. 2h), suggesting that *miro-1* is not required for MTX-2's mitochondrial localization. Hence, MTX-2 and MIRO-1 are localized to mitochondria independent of each other.

**MTX-2 directly binds to MIRO-1**. Next, we co-expressed tagged versions of MTX-2 and MIRO-1 in HEK293T cells and used co-immunoprecipitation assay to test if they interact with each other biochemically. This experiment showed that FLAG-MTX-2 could co-precipitate with HA-MIRO-1 and vice versa. DRP-1, a mitochondrial protein does not co-precipitate with MTX-2 or MIRO-1 (Supplementary Fig. 3f–i). To further test if MIRO-1 and MTX-2 bind directly to each other, we expressed tagged proteins in *E. coli*. GST pull-down experiments using bacterial extracts of tagged proteins showed that, compared with the GST control, GST-MIRO-1 could pull down Trx-MTX-2 but GST could not (Fig. 2l and Supplementary Fig. 3j), suggesting that MIRO-1 can directly binds to MTX-2.

To further characterize the MIRO-1/MTX-2 complex, we co-purified His-MIRO-1 and Trx-MTX-2 and examined the protein complex by analytical gel-filtration (Fig. 2k). Trx-MTX-2 and His-MIRO-1 were co-eluted as one peak, confirming that these two proteins directly bind to each other and form a complex in solution. Taken together, these data support a model in which MTX-2 and MIRO-1 form a complex on the mitochondrial outer membrane to mediate its intracellular transport (Fig. 2m).

**MTX-1 plays an important role in mitochondrial trafficking in neurons**. Mammalian metaxin1 and metaxin2 function as a complex on the mitochondria outer membrane[32]. The worm orthologue of metaxin1, MTX-1, could be co-immunoprecipitated by MTX-2 when they were co-expressed in HEK293T cells and vice versa (Supplementary Fig. 4a, b). To test if MTX-1 also functions in mitochondrial trafficking, we constructed a null mutant of *mtx-1* using CRISPR-Cas9. Interestingly, *mtx-1* mutants were devoid of mitochondria in posterior PVD dendrites but showed a slightly enhanced number of mitochondria in anterior

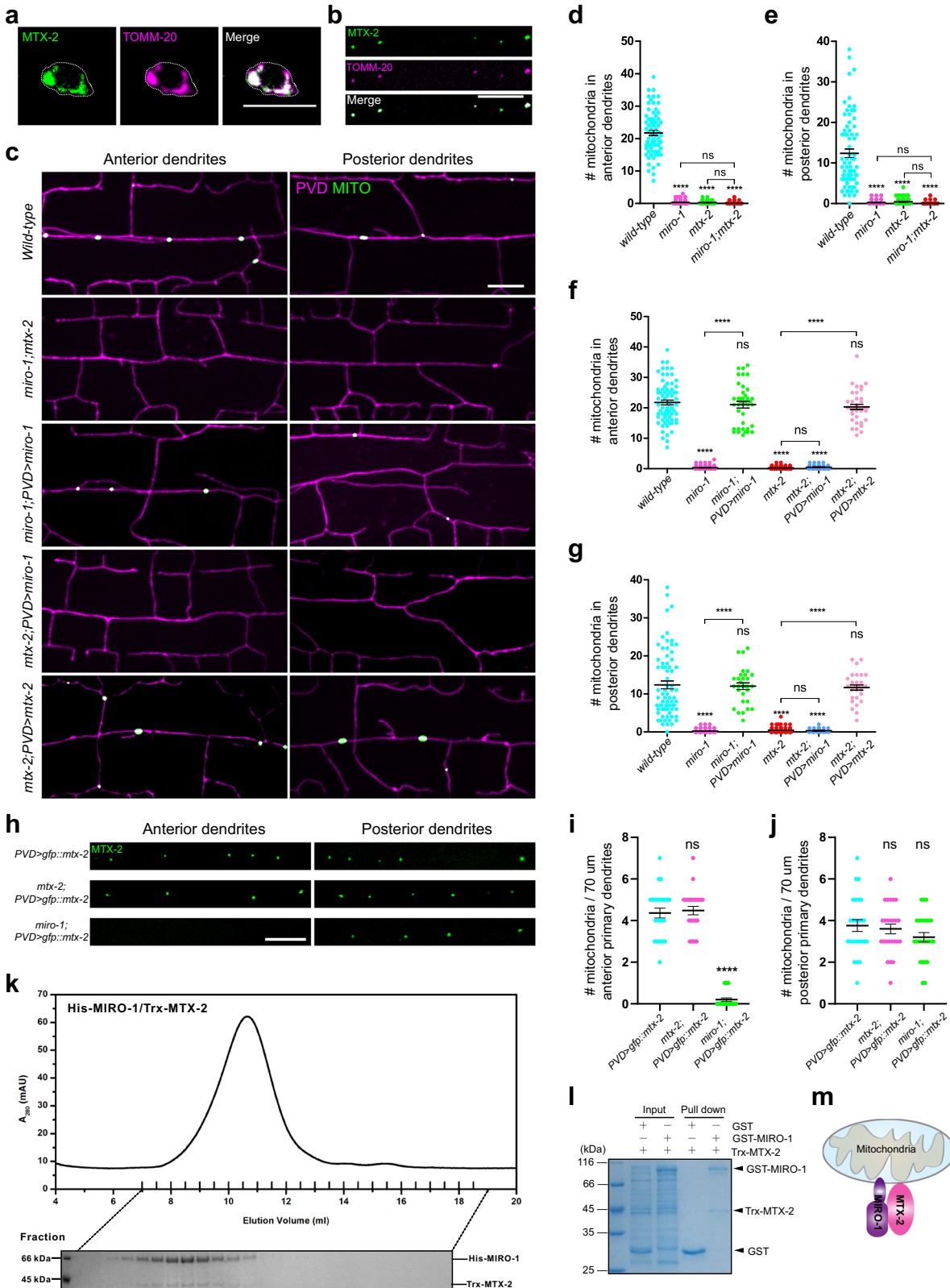

dendrites (Fig. 3a, b). Time-lapse imaging experiments showed that *mtx-1* mutants showed dynamic behavior defects only in posterior dendrites (Supplementary Fig. 4c, d). Western blot analysis showed that the level of MIRO-1 was not affected by the *mtx-1* mutation (Supplementary Fig. 3a–e).

The critical difference between anterior and posterior dendrites in PVD is that anterior dendrites contain exclusive minus-end-

out microtubule arrays (MTs), similar to other *C. elegans* dendrites, including the DA9 dendrite. However, posterior PVD dendrites contain plus-end-out microtubules similar to axons in other worm neurons (Fig. 3c)[28]. Consistently, *mtx-1* mutants showed reduced mitochondrial localization in the DA9 axon (plus-end-out MTs), but normal mitochondrial localization in the DA9 dendrite (minus-end-out MTs) (Fig. 3e–h). Together, these

**Fig. 2 MTX-2 interacts with MIRO-1 and functions in mitochondrial trafficking. a, b** Representative confocal images of a transgenic animal expressing GFP::MTX-2 (single copy) and TOMM-20::mCherry in PVD cell body (**a**) and dendrites (**b**). Scale bar: 10 μm. **c** Representative confocal images showing dendritic morphology and mitochondria distribution of PVD neuron in wild type, *miro-1 mtx-2* double mutants, and various PVD-specific rescue animals. Magenta: PVD > mCherry. Green: PVD > TOMM-20 (1-54AA)::GFP. Scale bar: 10 μm. **d–g** Quantification of mitochondria number in anterior and posterior dendrites in wild type and mutants with different genotype. Data are shown as mean ± SEM. One-way ANOVA with Tukey's multiple comparisons test (95% CI). $p > 0.05$, not significant, ****$p < 0.0001$. (exact *p* values and sample size: **d** wild-type vs. *miro-1*, $p < 0.0001$; wild-type vs. *mtx-2*, $p < 0.0001$; wild-type vs. *miro-1 mtx-2*, $p < 0.0001$; *miro-1* vs. *miro-1 mtx-2*, $p = 0.9800$; *mtx-2* vs. *miro-1 mtx-2*, $p = 0.9933$. $n = 72$ animals for each genotype. **e** wild-type vs. *miro-1*, $p < 0.0001$; wild-type vs. *mtx-2*, $p < 0.0001$; wild-type vs. *miro-1 mtx-2*, $p < 0.0001$; *miro-1* vs. *miro-1 mtx-2*, $p = 0.9959$; *mtx-2* vs. *miro-1 mtx-2*, $p = 0.9572$. $n = 72$ animals for each genotype. **f** wild-type vs. *miro-1*, $p < 0.0001$; wild-type vs. *miro-1* PV*D > miro-1*, $p = 0.9623$; wild-type vs. *mtx-2*, $p < 0.0001$; wild-type vs. *mtx-2* PVD > *miro-1*, $p < 0.0001$; wild-type vs. *mtx-2* PVD > *mtx-2*, $p = 0.5391$; *miro-1* vs. *miro-1* PVD > *miro-1*, $p < 0.0001$; *mtx-2* vs. *mtx-2* PVD > *mtx-2*, $p < 0.0001$; *mtx-2* vs. *mtx-2* PVD > *miro-1*, $p = 0.9998$. wild-type, $n = 72$; *miro-1*, $n = 72$; *miro-1* PVD> *miro-1*, $n = 39$; *mtx-2*, $n = 72$; *mtx-2* PVD> *miro-1*, $n = 39$; *mtx-2* PVD > *mtx-2*, $n = 39$ animals. **g** wild-type vs. *miro-1*, $p < 0.0001$; wild-type vs. *miro-1* PVD > *miro-1*, $p = 0.9993$; wild-type vs. *mtx-2*, $p < 0.0001$; wild-type vs. *mtx-2* PVD> *miro-1*, $p < 0.0001$; wild-type vs. *mtx-2* PVD > *mtx-2*, $p = 0.9848$; *miro-1* vs. *miro-1* PVD > *miro-1*, $p < 0.0001$; *mtx-2* vs. *mtx-2* PVD > *mtx-2*, $p < 0.0001$; *mtx-2* vs. *mtx-2* PVD > *miro-1*, $p > 0.9999$. wild-type, $n = 72$; *miro-1*, $n = 72$; *miro-1* PVD > *miro-1*, $n = 39$; *mtx-2*, $n = 72$; *mtx-2* PVD > *miro-1*, $n = 39$; *mtx-2* PVD > *mtx-2*, $n = 39$ animals). **h** Representative confocal images showing GFP::MTX-2 (single copy) in the anterior and posterior dendrites of wild type, *mtx-2*, and *miro-1* mutants. Scale bar: 10 μm. **i, j** Quantification of mitochondria number labeled by single-copy transgenic GFP::MTX-2 in wild type, *mtx-2*, and *miro-1* mutants in the anterior (**i**) and posterior (**j**) dendrites of PVD. Data are shown as mean ± SEM. One-way ANOVA with Tukey's multiple comparisons test (95% CI). $p > 0.05$, not significant, ****$p < 0.0001$. (exact *p* values and sample size: **i** PVD > *gfp::mtx-2* vs. *mtx-2* PVD > *gfp::mtx-2*, $p = 0.8914$; PVD > *gfp::mtx-2* vs. *miro-1* PVD > *gfp::mtx-2*, $p < 0.0001$; $n = 25$ animals for each genotype. **j** PVD > *gfp::mtx-2* vs. *mtx-2* PVD > *gfp::mtx-2*, $p = 0.8935$; PVD > *gfp::mtx-2* vs. *miro-1* PVD > *gfp::mtx-2*, $p = 0.2593$. $n = 25$ animals for each genotype). **k** Analytical gel-filtration analysis of the interaction between MIRO-1 and MTX-2. The proteins in each fraction were analyzed by SDS-PAGE with coomassie-blue staining. **l** GST pull-down assay of the interaction between MIRO-1 and MTX-2. GST was used as the control. $n = 3$ independent experiments. **m** A schematic diagram showing the MTX-2/MIRO-1 complex on mitochondria. Source data are provided as a Source Data file.

data prompt us to propose that MTX-1 is required for mitochondrial transport by the plus end-oriented motor kinesin, but not by dynein.

To test this hypothesis, we examined a partial loss of function allele of the kinesin-1 heavy chain/UNC-116, a conditional *unc-116* knockdown strain, as well as a loss-of-function allele of a kinesin light chain/*klc-1*. Consistent with our hypothesis, all three strains showed a reduced number of mitochondria in posterior dendrites (Fig. 3i, k). These strains also showed reduced mitochondria in anterior dendrites (Fig. 3i, j), which is likely due to the reversal of microtubule polarity in anterior dendrites of kinesin mutants. We have previously shown that in *unc-116* mutants, anterior PVD dendrites change their MTs polarity and contain mostly plus-end-out microtubules, while posterior dendrites preserve their plus-end-out MTs (Fig. 3c, d)[28].

Similar to the loss of mitochondria in PVD, the *unc-116* mutation completely blocked mitochondrial transport into both the DA9 axon and dendrite (Fig. 3e–h). This is not surprising since the same mutation also reverses MTs polarity in the DA9 dendrite to plus-end-out MTs[33]. *klc-1* mutants did not show a phenotype in DA9 (Fig. 3e–h), which might be due to the redundancy between KLC-1 and KLC-2. KLC-1 and KLC-2 are two closely related kinesin light chains sharing 46% sequence identity. *klc-2* null mutations cause larval lethality, precluding phenotypic analysis in neurons. However, we circumvented this complication by utilizing another weak allele of *klc-2*, which did show reduced mitochondrial number in the DA9 axon and dendrite (Fig. 3e–h). Together, these data suggest that MTX-1 is only required for kinesin-1-mediated mitochondrial transport, while MTX-2 and MIRO-1 are required for both kinesin- and dynein-mediated mitochondrial transport.

**MTX-1 functions together with MTX-2 and MIRO-1**. To understand the relationship between *mtx-1* and *mtx-2*, we ask if they rely on each other to be localized to mitochondria. A single-copy GFP::MTX-1 transgene showed a mitochondrial localization pattern in PVD (Fig. 4a). This localization was completely lost in the *mtx-2* mutant (Fig. 4a), suggesting that mitochondria-localized MTX-1 needs MTX-2 for targeting or stabilization. On the other hand, MTX-2 was localized to neuronal

mitochondria normally even in the absence of MTX-1 (Fig. 4b). These results are surprising because metaxin1 was shown to recruit metaxin2 to mitochondria in cultured mammalian cells[32]. Unlike in the *mtx-2* mutant, MTX-1 still localized to mitochondria in *miro-1* mutants (Fig. 4c).

Next, we created compound mutants between *mtx-1*, *mtx-2*, and *miro-1* to dissect the relationship between them. As predicted, *mtx-1 mtx-2* double mutants completely lacked mitochondria in both anterior and posterior dendrites in PVD and DA9 dendrites (Supplementary Figs. 4e–g and 2a–c). Similarly, *mtx-1 miro-1* double mutants showed a very similar phenotype to that of the *mtx-1 mtx-2* double mutants in both PVD and DA9 dendrites (Supplementary Figs. 4e–g and 2a–c). Since single mutations only caused a partial reduction of mitochondria in the DA9 axon, we used the enhancement of phenotypes to study the genetic interactions between these three genes. Interestingly, *mtx-1 mtx-2* double mutants did not show an enhanced phenotype compared to single mutants (Supplementary Fig. 2a–c), suggesting that they function in the same genetic pathway. Both *mtx-1 miro-1* and *mtx-2 miro-1* completely lacked mitochondria in the DA9 axon, indicating that MIRO-1 functions at least partially in parallel to MTX-1/2 (Supplementary Fig. 2a–c). The triple mutants also showed complete transport defects in the DA9 axon and dendrite (Supplementary Fig. 2a–c).

To further study this genetic interaction, we created transgenes that overexpress GFP::MTX-1 or GFP::MTX-2 in PVD. PVD > GFP::MTX-1 fully rescued the mitochondria phenotype in the *mtx-1* mutants (Fig. 4c, f, g), suggesting that the GFP fusion construct is functional and *mtx-1* functions cell autonomously to promote mitochondria transport. Overexpression of GFP::MTX-2 rescued the *mtx-1* mutant phenotype (Fig. 4b, d, e), while overexpression of GFP::MTX-1 could not rescue any aspect of the *mtx-2* mutant phenotype (Supplementary Fig. 4h). These results suggest that MTX-1 might help to increase the activity of MTX-2, which is essential for mitochondrial transport. Furthermore, overexpression of GFP::MTX-1 fully rescued the mitochondrial distribution phenotype of *miro-1* in the posterior dendrite but showed no rescue in the anterior dendrite (Fig. 4c, f, g). Since GFP::MTX-2 also rescued the posterior mitochondria phenotype of *miro-1* mutant, these data argue that the MTX-1/2 complex

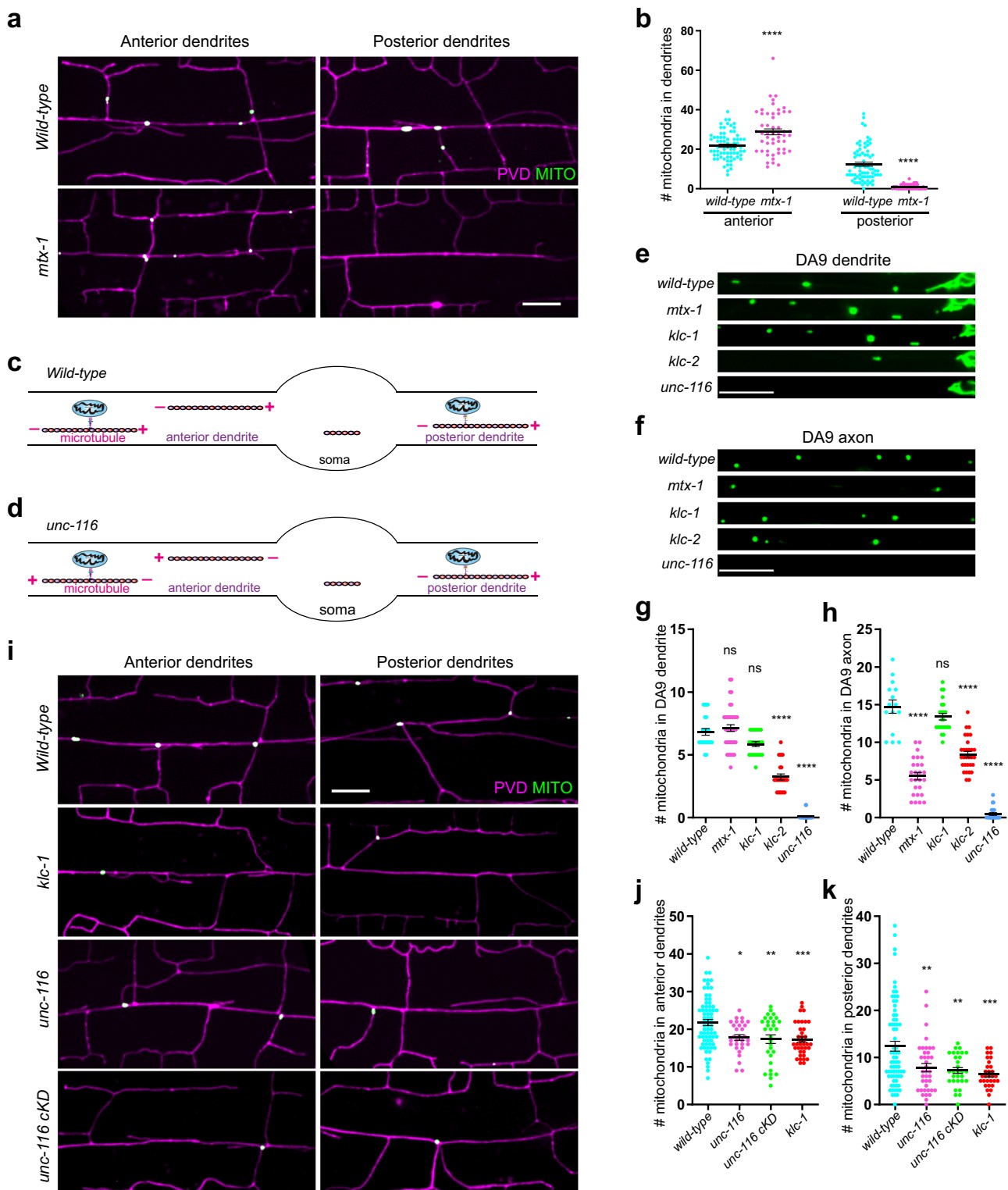

can substitute for MIRO-1 for mitochondrial transport by kinesin but MIRO-1 is absolutely required for trafficking towards the minus end of microtubules by dynein.

To further test if MIRO-1 interacts with Mtx1 and Mtx2 on mitochondria, we isolated mitochondria from HEK293T cells and performed the pull-down experiment using a Miro1 antibody. Both Mtx1 and Mtx2 could be detected in the immunoprecipitate, further supporting the notion that these three proteins interact with each other on the mitochondrial outer membrane (Fig. 4h).

**The MTX-2/MIRO-1/MTX-1 complex links KHC/UNC-116 through light chain KLC-1.** Miro is coupled to kinesin-1 either through direct binding to its kinesin heavy chain or indirectly through TRAKs[21,22]. Given that both MTX-2/MTX-1 and KLC-1 are important for mitochondria transport in PVD, we tested the direct binding between the MTX-1/MTX-2 complex, MIRO-1, and KLC-1, which functions as the cargo-binding subunit for kinesin-1 for many cargoes[34]. First, GST-MIRO-1 could pull-down the MTX-1/2 complex (formed by co-expression of Trx-

**Fig. 3 MTX-1 is required for kinesin-1 mediated mitochondrial trafficking in neurons. a** Representative confocal images showing dendritic morphology and mitochondrial distribution of PVD neuron in wild type and *mtx-1* mutant. Magenta: PVD > mCherry. Green: PVD > TOMM-20 (1-54AA)::GFP. Scale bar: 10 μm. **b** Quantification of mitochondria number in anterior and posterior dendrites of wild type and *mtx-1* mutant. Data are shown as mean ± SEM. Student's *t* test (95% CI). ****$p < 0.0001$. (exact *p* values and sample size: anterior, wild-type vs. *mtx-1*, $p < 0.0001$. wild-type, $n = 72$; *mtx-1*, $n = 52$ animals. posterior, wild-type vs. *mtx-1*, $p < 0.0001$. wild-type, $n = 72$; *mtx-1*, $n = 60$ animals). **c, d** Schematic diagrams showing the microtubule polarity in the anterior and posterior dendrites of PVD in wild type (**c**) and *unc-116* mutant (**d**). **e, f** Representative confocal images showing the mitochondrial distribution of DA9 neuron in wild type, *mtx-1, klc-1, klc-2,* and *unc-116* mutants. Green: DA9 > TOMM-20 (1-54AA)::GFP. Scale bar: 10 μm. **g, h** Quantification of mitochondria number in the dendrite and axon of DA9 in wild type, *mtx-1, klc-1, klc-2,* and *unc-116* mutants. Data are shown as mean ± SEM. One-way ANOVA with Tukey's multiple comparisons test (95% CI). $p > 0.05$, not significant, ****$p < 0.0001$. (exact *p* values and sample size: **g** wild-type vs. *mtx-1*, $p = 0.8855$; wild-type vs. *klc-1*, $p = 0.0777$; wild-type vs. *klc-2*, $p < 0.0001$; wild-type vs. *unc-116*, $p < 0.0001$. wild-type, $n = 23$; *mtx-1*, $n = 39$; *klc-1*, $n = 21$; *klc-2*, $n = 25$; *unc-116*, $n = 24$ animals. **h** wild-type vs. *mtx-1*, $p < 0.0001$; wild-type vs. *klc-1*, $p = 0.3830$; wild-type vs. *klc-2*, $p < 0.0001$; wild-type vs. *unc-116*, $p < 0.0001$. wild-type, $n = 15$; *mtx-1*, $n = 26$; *klc-1*, $n = 23$; *klc-2*, $n = 30$; *unc-116*, $n = 23$ animals). **i** Representative confocal images showing dendritic morphology and mitochondria distribution of PVD neuron in wild type, *klc-1, unc-116* mutants, and PVD-specific RNAi of *unc-116* animals. Magenta: PVD > mCherry. Green: PVD > TOMM-20 (1-54AA)::GFP. Scale bar: 10 μm. **j, k** Quantification of mitochondria number in the anterior (**j**) and posterior (**k**) dendrites of wild type and mutants. Data are shown as mean ± SEM. One-way ANOVA with Tukey's multiple comparisons test (95% CI). $p > 0.05$, not significant, ***$p < 0.001$, **$p < 0.01$, *$p < 0.05$. (exact *p* values and sample size: **j** wild-type vs. *unc-116*, $p = 0.0108$; wild-type vs. *unc-116* cKD, $p = 0.0036$; wild-type vs. *klc-1*, $p = 0.0009$. wild-type, $n = 72$; *unc-116*, $n = 30$; *unc-116* cKD, $n = 30$; *klc-1*, $n = 38$ animals. **k** wild-type vs. *unc-116*, $p = 0.0040$; wild-type vs. *unc-116* cKD, $p = 0.0023$; wild-type vs. *klc-1*, $p = 0.0003$. wild-type, $n = 72$; *unc-116*, $n = 38$; *unc-116* cKD, $n = 30$; *klc-1*, $n = 30$ animals). Source data are provided as a Source Data file.

MTX-1 and Trx-MTX2) in a roughly stoichiometric ratio from mixed cell lysates (Fig. 4j and Supplementary Fig. 3j). To further prove that these three proteins form a complex, we used analytical gel-filtration analysis to show that His-MIRO-1 could be co-eluted with the MTX-1/2 complex (Trx-MTX-1 and Trx-MTX-2) in a single major peak that contained all three proteins (Fig. 4i). These results strongly suggest that MIRO-1 binds to the MTX-1/2 complex directly. Moreover, in the additional pull-down assay with changing the Trx-MTX-1 dosage, no visible difference in the amount of Trx-MTX-1 was observed in the complex, confirming that these three proteins indeed exist in the same complex (Supplementary Fig. 4i).

Next, with the same type of pull-down experiments and gel-filtration analysis, we showed that the MTX-1/2 complex could directly bind to the TPR domain of KLC-1, which is the primary cargo-binding domain of kinesin light chains, and the three proteins (Trx-MTX-1, Trx-MTX-2, and His-KLC-1-TPR) could be co-eluted as one major peak in gel-filtration (Fig. 4k, l and Supplementary Fig. 3j). Lastly, GST-MIRO-1 could pull-down the MTX-1/MTX-2 complex together with KLC-1-TPR and all four proteins (Trx-MTX-1, Trx-MTX-2, His-MIRO-1, and His-KLC-1-TPR) could be co-eluted together as a single peak in our gel-filtration analysis (Fig. 4m, n and Supplementary Fig. 3j). These biochemical data using purified proteins demonstrate that the metaxin complex can form a complex with MIRO-1 and a kinesin light chain. Together with the strong loss-of-function phenotype of these genes, we propose that the MIRO-1/metaxins tri-protein complex functions as an essential adaptor to couple mitochondria to kinesin-1 for trafficking to neurites (Fig. 4o).

**The MTX-2/MIRO-1/TRAK-1 complex mediates mitochondrial transport by dynein.** To further understand mitochondrial trafficking to the anterior dendrite, which contains predominantly minus-end-out microtubules, we specifically ask whether dynein is involved and how metaxins link to dynein. The dynein complex is composed of a heavy chain, a light chain, a light intermediate chain, and intermediate chains. We obtained a *dli-1* (dynein light intermediate chain) mutant in our screen, which blocked mitochondrial localization in PVD anterior dendrites but had no effect on mitochondria in posterior dendrites (Fig. 5a–c). In mammalian cells, Miro and Milton/TRAK also couple mitochondria to dynein. We then generated a null mutant of *trak-1*, the sole orthologue of Milton/TRAK in worm, and found that this mutation showed a very similar effect as the *dli-1*

mutation in the distribution of mitochondria in anterior dendrites (Fig. 5a–c). This result suggests that while MTX-1 is specifically required for plus-end-out movements, *trak-1* is required for minus-end-out movement of mitochondria. Our finding is consistent with experiments in DA9 (Fig. 5d–f), showing that *trak-1* only blocked movement of mitochondria to the dendrite but not the axon. Time-lapse imaging of *trak-1* showed that "extension" events were reduced and no "transport" events were observed during a total of 14.5 h of recordings in the anterior dendrite, while it was normal in the posterior dendrite, indicating it functions solely in the trafficking of mitochondria to the anterior dendrite (Supplementary Fig. 5j). These data suggest that TRAK-1 in *C. elegans* is only required for dynein-mediated mitochondrial transport, but not required for kinesin-1-mediated transport, while MTX-1 does the opposite. Consistent with this model, *trak-1 mtx-1* double mutants nearly completely lost mitochondria in both PVD anterior and posterior dendrites (Supplementary Fig. 5a–c). In DA9 axons, a *trak-1* mutation could not enhance *mtx-1* or *mtx-2* mitochondria phenotype, which is due to reduced kinesin-1-mediated transport (Supplementary Fig. 5d, f).

In mammalian and *Drosophila* neurons, TRAK bridges Miro and motor proteins. To test if *C. elegans* TRAK-1 performs a similar function, we performed co-immunoprecipitation experiments by co-expressing HA-MIRO-1 and FLAG-TRAK-1 in HEK293T cells. We found that HA-MIRO-1 could indeed be efficiently co-precipitated by FLAG-TRAK-1 and vice versa (Supplementary Fig. 5g, h), indicating that this interaction is conserved. Since MTX-2 is also required for trafficking of mitochondria to anterior dendrites, we also tested potential interaction between TRAK-1 and MTX-2 with the same method. We found that HA-MTX-2 could be co-precipitated by FLAG-TRAK-1 and vice versa (Supplementary Fig. 5g, i), albeit not as strong as MIRO-1. This interaction is likely to be specific because the FLAG tag alone or FLAG-DRP-1 did not pull-down any HA-MTX-2 or HA-TRAK-1 (Supplementary Fig. 3f, g). Together, these data suggest that MTX-2, MIRO-1, and TRAK-1 form a complex to link mitochondria to the dynein motor (Fig. 5k). However, the unstable nature of TRAK-1 as a purified protein prevented us from testing this hypothesis using analytical gel-filtration analysis and GST pull-down assays.

Overexpression of MTX-1 or MTX-2 in PVD completely rescued mitochondrial localization in the posterior dendrite of the *miro-1* mutant, but showed no rescue in the anterior dendrite

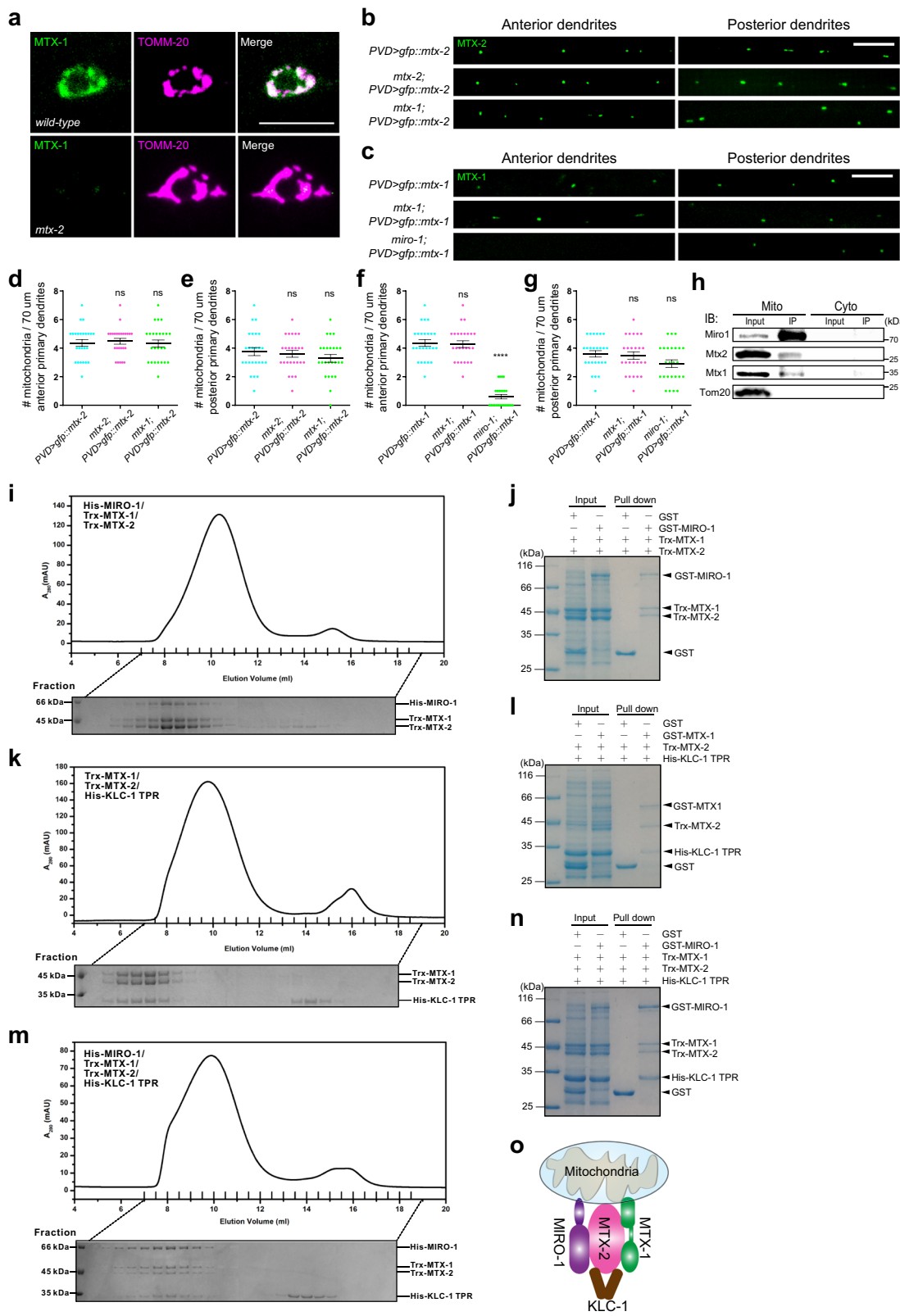

(Figs. 2h–j and 4c, f, g). This result suggested that the metaxin complex alone could mediate kinesin coupling, while MIRO-1 is indispensable for linking mitochondria to dynein. To further test this idea, we overexpressed *mtx-1*, *mtx-2*, and *miro-1* separately in *trak-1* mutants and found that none of them could rescue the *trak-1* mutant phenotype (Fig. 5a–c, g–j), confirming that MIRO-

1/TRAK-1 are indispensable for coupling mitochondria to dynein.

**Impairment of mitochondrial trafficking leads to dendrite degeneration.** Mitochondria play a pivotal role in neuronal function by producing ATP and buffering intracellular calcium.

**Fig. 4 The MTX-2/MIRO-1/MTX-1 complex links KHC/UNC-116 through light chain KLC-1. a** Representative confocal images showing GFP::MTX-1 (single copy) and TOMM-20::mCherry in PVD cell body of wild type and *mtx-2* mutant. Scale bar: 10 μm. **b, c** Representative confocal images of GFP::MTX-2 (**b**) and GFP::MTX-1 (**c**) in wild type, *mtx-2*, *mtx-1*, or *miro-1* mutants in the anterior and posterior dendrites of PVD. Scale bar: 10 μm. **d–g** Quantification of mitochondria number by overexpressing GFP::MTX-2 and GFP::MTX-1 in wild type, *mtx-2*, *mtx-1*, or *miro-1* mutants in anterior and posterior dendrites of PVD. Data are shown as mean ± SEM. One-way ANOVA with Tukey's multiple comparisons test (95% CI). $p > 0.05$, not significant, ****$p < 0.0001$. (exact *p* values and sample size: **d** PVD > *gfp::mtx-2* vs. *mtx-2* PVD > *gfp::mtx-2*, $p = 0.9280$; PVD > *gfp::mtx-2* vs. *mtx-1* PVD > *gfp::mtx-2*, $p = 0.9917$; $n = 25$ animals for each genotype. **e** PVD > *gfp::mtx-2* vs. *mtx-2* PVD > *gfp::mtx-2*, $p = 0.9040$; PVD > *gfp::mtx-2* vs. *mtx-1* PVD > *gfp::mtx-2*, $p = 0.4083$; $n = 25$ animals for each genotype. **f** PVD > *gfp::mtx-1* vs. *mtx-1* PVD > *gfp::mtx-1*, $p = 0.9615$; PVD > *gfp::mtx-1* vs. *miro-1* PVD > *gfp::mtx-1*, $p < 0.0001$; $n = 25$ animals for each genotype. **g** PVD > *gfp::mtx-1* vs. *mtx-1* PVD > *gfp::mtx-1*, $p = 0.9372$; PVD > *gfp::mtx-1* vs. *miro-1* PVD > *gfp::mtx-1*, $p = 0.1338$; $n = 25$ animals for each genotype). **h** Isolated mitochondrial lysates of HEK293T cells were immunoprecipitated with ANTI-Miro1 antibody, followed by western blot analysis with Miro1, Mtx2, Mtx1, and Tom20 antibodies. $n = 3$ independent experiments. In the experiment, the input was 2.5% of the total isolated mitochondrial lysates, and the IP was 25% of the total elution. $n = 3$ independent experiments. The cytosol fraction was included as a control. **i** Analytical gel-filtration analysis of the interaction between MIRO-1 and the MTX-1/MTX-2 complex. **j** GST pull-down assay between MIRO-1 and the MTX-1/MTX-2 complex. **k** Analytical gel-filtration analysis of the interaction between the MTX-1/MTX-2 complex and the TPR domain of KLC-1 (KLC-1-TPR). **l** GST pull-down assay of the interaction between the MTX-1/MTX-2 complex and KLC-1-TPR. **m** Analytical gel-filtration analysis of the formation of the MIRO-1/MTX-2/MTX-1/KLC-1-TPR complex in solution. **n** GST pull-down assay of the MIRO-1/MTX-2/MTX-1/KLC-1-TPR complex. $n = 3$ independent experiments. **o** A schematic diagram showing the putative molecular complex containing MTX-2, MTX-1, MIRO-1, and KLC-1 on mitochondria. Source data are provided as a Source Data file.

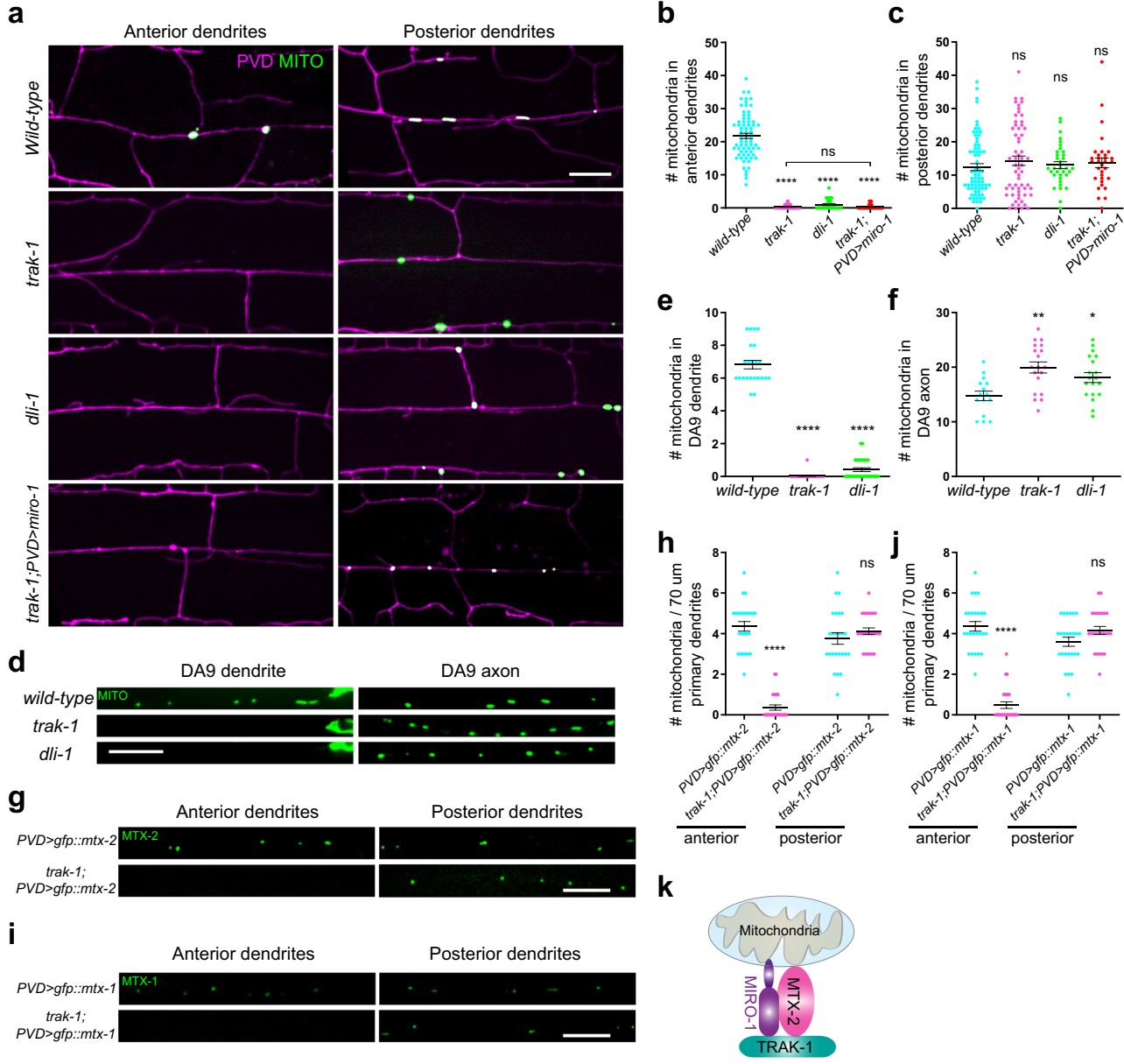

**Fig. 5 The MTX-2/MIRO-1/TRAK-1 complex mediates dynein-based mitochondria transport. a** Representative confocal images showing dendritic morphology and mitochondrial distribution of PVD neuron in wild type, *trak-1*, *dli-1* mutants, and *trak-1* PVD > *miro-1* animals. Magenta: PVD dendrites. Green: PVD > TOMM-20 (1-54AA)::GFP. Scale bar 10 μm. **b**, **c** Quantification of mitochondria number in the anterior (**b**) and posterior (**c**) dendrites of PVD in wild type, *trak-1*, *dli-1* mutants, and *trak-1* PVD > *miro-1* animals. Data are shown as mean ± SEM. One-way ANOVA with Tukey's multiple comparisons test (95% CI). $p > 0.05$, not significant, ****$p < 0.0001$. (exact $p$ values and sample size: **b** wild-type vs. *trak-1*, $p < 0.0001$; wild-type vs. *dli-1*, $p < 0.0001$; wild-type vs. *trak-1* PVD > *miro-1*, $p < 0.0001$. wild-type, $n = 72$; *trak-1*, $n = 72$; *dli-1*, $n = 35$; *trak-1* PVD > *miro-1*, $n = 35$ animals. **c** wild-type vs. *trak-1*, $p = 0.6263$; wild-type vs. *dli-1*, $p = 0.9880$; wild-type vs. *trak-1* PVD > *miro-1*, $p = 0.9015$. wild-type, $n = 72$; *trak-1*, $n = 60$; *dli-1*, $n = 35$; *trak-1* PVD > *miro-1*, $n = 32$ animals). **d** Representative confocal images showing mitochondria distribution of DA9 neuron in wild type, *trak-1*, and *dli-1* mutants. Green: DA9 > TOMM-20 (1-54AA)::GFP. Scale bar: 10 μm. **e**, **f** Quantification of mitochondria number in dendrite and axon of DA9 in wild type, *trak-1*, and *dli-1* mutants. Data are shown as mean ± SEM. One-way ANOVA with Tukey's multiple comparisons test (95% CI). $p > 0.05$, not significant, ****$p < 0.0001$, **$p < 0.01$, *$p < 0.05$. (exact $p$ values and sample size: **e** wild-type vs. *trak-1*, $p < 0.0001$; wild-type vs. *dli-1*, $p < 0.0001$. wild-type, $n = 23$; *trak-1*, $n = 28$; *dli-1*, $n = 29$ animals. **f** wild-type vs. *trak-1*, $p = 0.0012$; *wild-type* vs. *dli-1*, $p = 0.0458$. *wild-type*, $n = 15$; *trak-1*, $n = 19$; *dli-1*, $n = 19$ animals). **g** Representative confocal images showing GFP::MTX-2 (single copy) in wild type and *trak-1* mutant in the anterior and posterior dendrites of PVD. Scale bar: 10 μm. **h** Quantification of mitochondria number by overexpressing GFP::MTX-2 in wild type and *trak-1* mutant in anterior and posterior dendrites of PVD. Data are shown as mean ± SEM. Student's $t$ test (95% CI). $p > 0.05$, not significant, ****$p < 0.0001$. (exact $p$ values and sample size: anterior, PVD > *gfp::mtx-2* vs. *trak-1* PVD > *gfp::mtx-2*, $p < 0.0001$, $n = 25$ animals for each genotype. posterior, PVD > *gfp::mtx-2* vs. *trak-1* PVD > *gfp::mtx-2*, $p = 0.2802$, $n = 25$ animals for each genotype). **i** Representative confocal images showing single-copy transgenic GFP::MTX-1 in wild type and *trak-1* mutant in anterior and posterior dendrites of PVD. Scale bar: 10 μm. **j** Quantification of mitochondria number of single-copy transgenic GFP::MTX-1 in wild type and *trak-1* mutant in anterior and posterior dendrites of PVD. Data are shown as mean ± SEM. Student's $t$ test (95% CI). $p > 0.05$, not significant, ****$p < 0.0001$. (exact $p$ values and sample size: anterior, PVD > *gfp::mtx-1* vs. *trak-1* PVD > *gfp::mtx-1*, $p < 0.0001$, $n = 25$ animals for each genotype. posterior, PVD > *gfp::mtx-1* vs. *trak-1* PVD > *gfp::mtx-1*, $p = 0.0665$, $n = 25$ animals for each genotype). **k** A schematic diagram showing the putative complex of MTX-2, MIRO-1, and TRAK-1 on mitochondria. Source data are provided as a Source Data file.

To understand the physiological significance of dendritic mitochondria, we examined the effect of mitochondrial trafficking mutants on PVD dendritic integrity. Strikingly, *miro-1*, *mtx-2*, and *trak-1* mutants, which all lacked mitochondria in the anterior dendrite, displayed age-dependent degeneration of distal secondary and quaternary anterior dendrites (Fig. 6a–d). Minor morphological phenotypes were observed in day 1 adults, indicating that dendritic outgrowth is largely normal. However, maintenance of the PVD dendrites appears to require local mitochondria in the dendrites. Degeneration occurred only in anterior dendrites without affecting posterior dendrites, correlating with the specific loss of mitochondria from anterior dendrites in these three mutants. Consistently, we did not observe any degeneration of anterior dendrites in the *mtx-1* mutant (Fig. 6b–d). We also did not observe any degeneration of posterior dendrites in the *mtx-1* mutant or any other mutants. We speculate that the short length of the posterior dendrite allows ATP or other mitochondria-derived molecules to efficiently diffuse from the soma to dendrites, and therefore posterior dendrites are less susceptible to mitochondrial trafficking defects. Consistent with this notion, even the degeneration of the anterior dendrite only occurred in the distal dendrite covering roughly 2/3 of the normal dendritic arbor. Together, these data suggest that dendritic mitochondria play critical roles in locally maintaining dendrite morphology during the aging process.

**Loss of the metaxin complex impairs mitochondria transport in human neurons.** Thus far, we have shown that MTX-1 and MTX-2 form a vital adaptor complex to mediate mitochondrial transport into the dendrites and axons of neurons in *C. elegans*. To test if their functions are conserved in human neurons, we knocked down *Mtx1*, *Mtx2*, *Trak1*, and *Trak2* using RNAi in induced pluripotent stem cell (iPSC)-derived neurons from a healthy human subject (Fig. 7a and Supplementary Fig. 6a–d). We transfected those neurons with mito-DsRed to label mitochondria and recorded live mitochondria movements in axons. Consistent with previous reports, *Trak1*, and *Trak2* knockdown reduced mitochondrial movements and increased stop frequency in both anterograde and retrograde directions (Fig. 7a–d), suggesting that the TRAK adaptors are important for both kinesin and dynein-mediated mitochondria transport. Notably,

knockdown of *Mtx1* or *Mtx2* significantly compromised mitochondrial movement in both directions, as evidenced by the significant reduction in the percentage of time that mitochondria spent in motion, the increase in stop frequency, and the decrease of turn back frequency (Fig. 7a–e). These results demonstrate that the function of metaxin in mitochondria trafficking is conserved in human neurons.

In summary, we have identified an evolutionarily conserved protein complex that acts as a crucial adaptor necessary for linking mitochondria with microtubule motor proteins. This complex functions together with the known MIRO-1 adaptor protein and likely generates two separate adaptor complexes to accommodate movement by kinesin and dynein motors.

## Discussion

To better understand the molecular mechanisms of mitochondrial trafficking in neurons, we established an in vivo system using the PVD sensory neuron in *C. elegans*. Several features of PVD are distinct from existing *Drosophila* and mammalian neuronal culture systems, both of which have made important contributions to our understanding of mitochondrial trafficking. First, the clear mitochondrial distribution within the elaborate PVD dendritic arbors makes it possible to visualize and study mitochondria trafficking in dendrites with single-cell resolution. Second, the distinct microtubule polarity in anterior and posterior dendrites is useful for distinguishing specific motor programs that are affected by mutations. Third, our ability to perform dynamic imaging of mitochondrial "extension", "fission" and "transport" from soma to neurites helps to pinpoint the precise molecular defects in mutants. Last, it appears that mitochondrial trafficking mutants that we have isolated so far do not affect organismal viability, making the isolation of homozygous mutants possible. In a forward visual genetic screen covering 4000 haploid genomes, we recovered alleles of several known genes for mitochondrial trafficking and fission including *miro-1*, *dli-1*, and *drp-1* (dynamin-related protein 1).

In the literature, Miro is the established crucial link between motor adaptor proteins and mitochondria, although other factors such as Syntabulin, FEZ1, RanBP2 have also been reported to link KIF5 to mitochondria[35–37]. Evolutionarily conserved interactions between Miro and Milton/TRAKs are important for both kinesin-

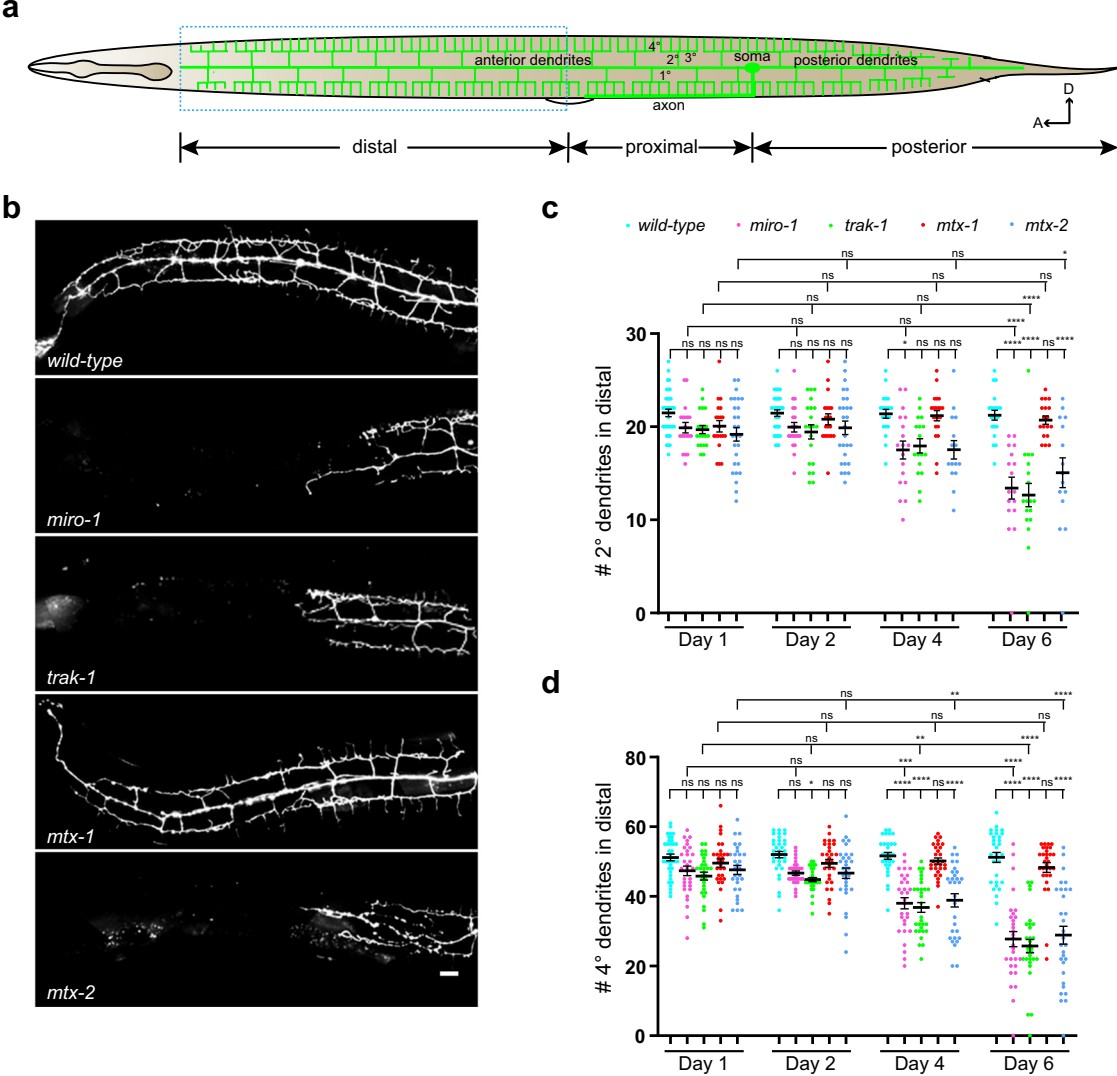

**Fig. 6 Impairment of mitochondrial trafficking leads to dendrite degeneration. a** A schematic diagram showing the distal, proximal, and posterior dendrites of PVD. Anterior is to the left and dorsal is up. **b** Confocal images of PVD > GFP in wild type, *miro-1, trak-1, mtx-1,* and *mtx-2* mutants at Day 6. Scale bar: 10 μm. **c, d** Quantification of the number of 2° and 4° in the distal dendrite in wild type, *miro-1, trak-1, mtx-1,* and *mtx-2* mutants at Day 1, 2, 4, and 6. Data are shown as mean ± SEM. One-way ANOVA with Tukey's multiple comparisons test (95% CI). $p > 0.05$, not significant, ****$p < 0.0001$, ***$p < 0.001$, **$p < 0.01$, *$p < 0.05$. (exact p values and sample size: **c** wild-type D1 vs. *miro-1* D1, $p = 0.9764$; wild-type D1 vs. *trak-1* D1, $p = 0.9176$; wild-type D1 vs. *mtx-1* D1, $p = 0.9908$; wild-type D1 vs. *mtx-2* D1, $p = 0.3889$; wild-type D2 vs. *miro-1* D2, $p = 0.9816$; wild-type D2 vs. *trak-1* D2, $p = 0.7995$; wild-type D2 vs. *mtx-1* D2, $p > 0.9999$; wild-type D2 vs. *mtx-2* D2, $p = 0.9419$; wild-type D4 vs. *miro-1* D4, $p = 0.0319$; wild-type D4 vs. *trak-1* D4, $p = 0.1314$; wild-type D4 vs. *mtx-1* D4, $p > 0.9999$; wild-type D4 vs. *mtx-2* D4, $p = 0.0627$; wild-type D6 vs. *miro-1* D6, $p < 0.0001$; wild-type D6 vs. *trak-1* D6, $p < 0.0001$; wild-type D6 vs. *mtx-1* D6, $p > 0.9999$; wild-type D6 vs. *mtx-2* D6, $p < 0.0001$; *miro-1* D1 vs. *miro-1* D2, $p > 0.9999$; *miro-1* D1 vs. *miro-1* D4, $p = 0.7763$; *miro-1* D1 vs. *miro-1* D6, $p < 0.0001$; *trak-1* D1 vs. *trak-1* D2, $p > 0.9999$; *trak-1* D1 vs. *trak-1* D4, $p = 0.9856$; *trak-1* D1 vs. *trak-1* D6, $p < 0.0001$; *mtx-1* D1 vs. *mtx-1* D2, $p > 0.9999$; *mtx-1* D1 vs. *mtx-1* D4, $p = 0.9999$; *mtx-1* D1 vs. *mtx-1* D6, $p > 0.9999$; *mtx-2* D1 vs. *mtx-2* D2, $p > 0.9999$; *mtx-2* D1 vs. *mtx-2* D4, $p = 0.9907$; *mtx-2* D1 vs. *mtx-2* D6, $p = 0.0146$. wild-type D1, $n = 35$; *miro-1* D1, $n = 19$; *trak-1* D1, $n = 20$; *mtx-1* D1, $n = 20$; *mtx-2* D1, $n = 27$; wild-type D2, $n = 35$; *miro-1* D2, $n = 21$; *trak-1* D2, $n = 20$; *mtx-1* D2, $n = 20$; *mtx-2* D2, $n = 27$; wild-type D4, $n = 21$; *miro-1* D4, $n = 18$; *trak-1* D4, $n = 17$; *mtx-1* D4, $n = 21$; *mtx-2* D4, $n = 15$; wild-type D6, $n = 21$; *miro-1* D6, $n = 17$; *trak-1* D6, $n = 18$; *mtx-1* D6, $n = 18$; *mtx-2* D6, $n = 15$ animals. **d** wild-type D1 vs. *miro-1* D1, $p = 0.9160$; wild-type D1 vs. *trak-1* D1, $p = 0.4083$; wild-type D1 vs. *mtx-1* D1, $p > 0.9999$; wild-type D1 vs. *mtx-2* D1, $p = 0.9562$; wild-type D2 vs. *miro-1* D2, $p = 0.4225$; wild-type D2 vs. *trak-1* D2, $p = 0.0355$; wild-type D2 vs. *mtx-2* D2, $p = 0.9994$; wild-type D2 vs. *mtx-2* D2, $p = 0.4100$; wild-type D4 vs. *miro-1* D4, $p < 0.0001$; wild-type D4 vs. *trak-1* D4, $p < 0.0001$; wild-type D4 vs. *mtx-1* D4, $p > 0.9999$; wild-type D4 vs. *mtx-2* D4, $p < 0.0001$; wild-type D6 vs. *miro-1* D6, $p < 0.0001$; wild-type D6 vs. *trak-1* D6, $p < 0.0001$; wild-type D6 vs. *mtx-1* D6, $p = 0.9972$; wild-type D6 vs. *mtx-2* D6, $p < 0.0001$; *miro-1* D1 vs. *miro-1* D2, $p > 0.9999$; *miro-1* D1 vs. *miro-1* D4, $p = 0.0010$; *miro-1* D1 vs. *miro-1* D6, $p < 0.0001$; *trak-1* D1 vs. *trak-1* D2, $p > 0.9999$; *trak-1* D1 vs. *trak-1* D4, $p = 0.0018$; *trak-1* D1 vs. *trak-1* D6, $p < 0.0001$; *mtx-1* D1 vs. *mtx-1* D2, $p > 0.9999$; *mtx-1* D1 vs. *mtx-1* D4, $p > 0.9999$; *mtx-1* D1 vs. *mtx-1* D6, $p > 0.9999$; *mtx-2* D1 vs. *mtx-2* D2, $p > 0.9999$; *mtx-2* D1 vs. *mtx-2* D4, $p = 0.0032$; *mtx-2* D1 vs. *mtx-2* D6, $p < 0.0001$. wild-type D1, $n = 35$; *miro-1* D1, $n = 30$; *trak-1* D1, $n = 30$; *mtx-1* D1, $n = 30$; *mtx-2* D1, $n = 30$; wild-type D2, $n = 35$; *miro-1* D2, $n = 30$; *trak-1* D2, $n = 30$; *mtx-1* D2, $n = 30$; *mtx-2* D2, $n = 30$; wild-type D4, $n = 30$; *miro-1* D4, $n = 30$; *trak-1* D4, $n = 30$; *mtx-1* D4, $n = 30$; *mtx-2* D4, $n = 30$; wild-type D6, $n = 30$; *miro-1* D6, $n = 30$; *trak-1* D6, $n = 30$; *mtx-1* D6, $n = 30$; *mtx-2* D6, $n = 30$ animals). Source data are provided as a Source Data file.

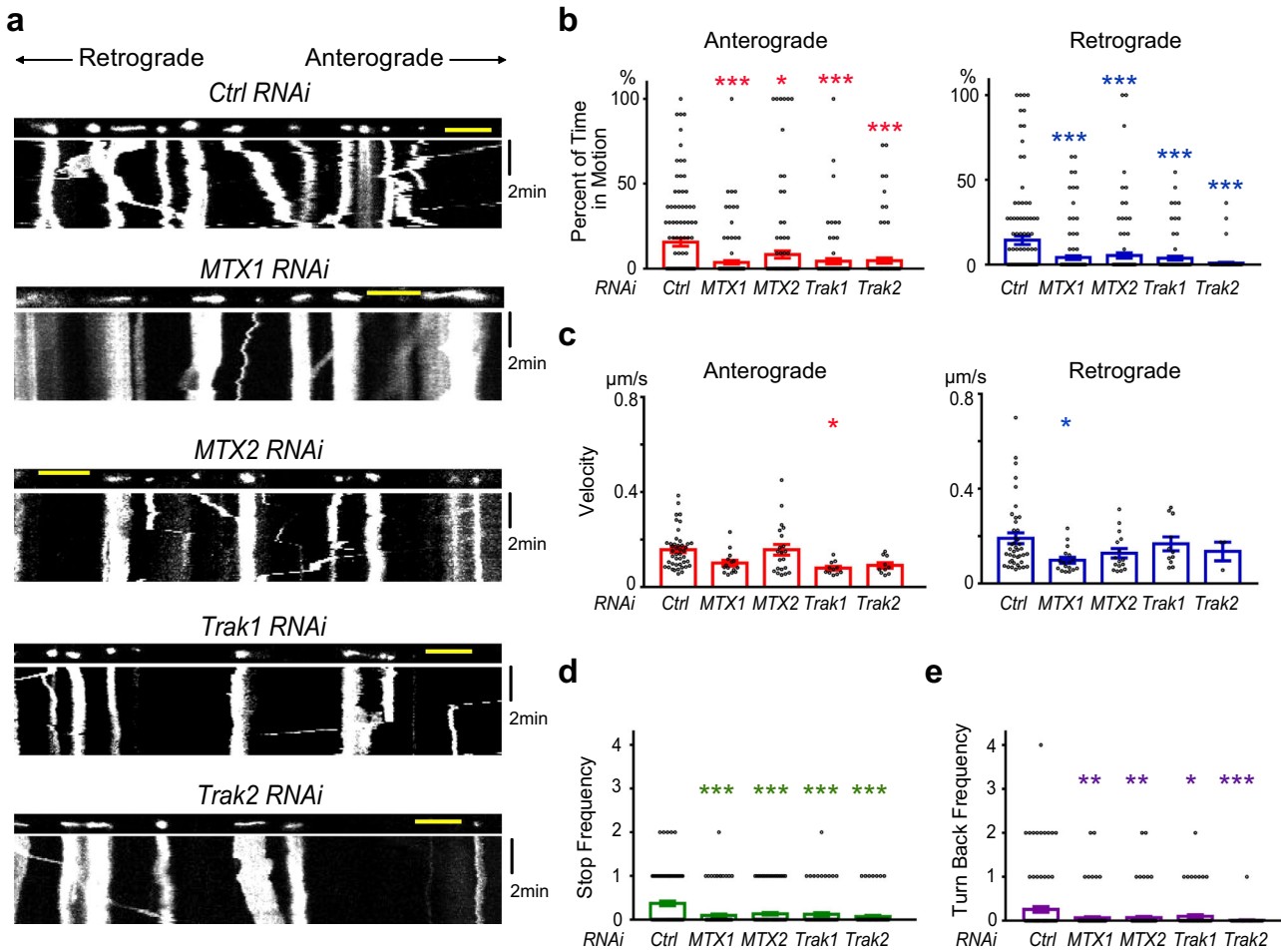

**Fig. 7 Mitochondria trafficking analyses in human iPSC-derived axons. a** Mitochondrial movement in representative axons transfected with mito-dsRed. The first frame of each live-imaging series is shown above a kymograph generated from the movie. The *x*-axis is mitochondrial position and the *y*-axis corresponds to time (moving from top to bottom). Scale bars: 10 μm. **b**–**e** From kymographs as in (**a**), Percent of Time each mitochondrion is in motion (**b**), Average Velocity (**c**), Stop Frequency (**d**), or Turn Back Frequency (**e**) is quantified. Percent of Time in Motion is calculated by the number of instantaneous velocities that are not zero divided by the total number of instantaneous velocities (including both zero and non-zero) for a mitochondrion; Average Velocity is defined as the average velocity of all instantaneous velocities that are not zero for a mitochondrion; Stop Frequency is defined as the frequency per mitochondrion that the instantaneous velocity changes to zero; Turn Back Frequency is defined as the frequency per mitochondrion that the direction of instantaneous velocity reverses. Data are shown as mean ± SEM. One-way ANOVA with Tukey's multiple comparisons test (95% CI). \**p* < 0.05, \*\**p* < 0.01, \*\*\**p* < 0.001. (exact *p* values and sample size: **b** anterograde, *p* < 0.0001 (*MTX1*); *p* = 0.0294 (*MTX2*); *p* = 0.0003 (*Trak1*); *p* = 0.0005 (*Trak2*). retrograde, *p* < 0.0001 (*MTX1*); *p* = 0.0003 (*MTX2*); *p* < 0.0001 (*Trak1*); *p* < 0.0001 (*Trak2*). **c** anterograde, *p* = 0.0950 (*MTX1*); *p* > 0.9999 (*MTX2*); *p* = 0.0231 (*Trak1*); *p* = 0.0947 (*Trak2*). retrograde, *p* = 0.0474 (*MTX1*); *p* = 0.3399 (*MTX2*); *p* = 0.9733 (*Trak1*); *p* = 0.9254 (*Trak2*). **d** *p* < 0.0001 (*MTX1*); *p* < 0.0001 (*MTX2*); *p* < 0.0001 (*Trak1*); *p* < 0.0001 (*Trak2*). **e** *p* = 0.0015 (*MTX1*); *p* = 0.0033 (*MTX2*); *p* = 0.0425 (*Trak1*); *p* < 0.0001 (*Trak2*). *n* = 113 (control RNAi), 138 (*MTX1* RNAi), 125 (*MTX2* RNAi), 90 (*Trak1* RNAi), 95 (*Trak2* RNAi) mitochondria (**b, d, e**); 44 (control RNAi), 15 (*MTX1* RNAi), 21 (*MTX2* RNAi), 11 (*Trak1* RNAi), 10 (*Trak2* RNAi) (**c**, anterograde); and 41 (control RNAi), 17 (*MTX1* RNAi), 16 (*MTX2* RNAi), 11 (*Trak1* RNAi), 3 (*Trak2* RNAi) (**c**, retrograde) from 12 (control RNAi), 16 (*MTX1* RNAi), 14 (*MTX2* RNAi), 10 (*Trak1* RNAi), 11 (*Trak2* RNAi) axons from four independent transfections. Source data are provided as a Source Data file.

and dynein-mediated mitochondrial transport. Mammalian cells have two TRAKs, TRAK1 and TRAK2, which are enriched in axon and dendrite, respectively. In the present study, we expand on the existing molecular mechanisms of mitochondrial trafficking in neurons by showing that the metaxin complex functions as an adaptor on the mitochondrial outer membrane to recruit kinesin and dynein. Our genetic data indicate that MTX-1 and MTX-2 play essential, but distinct, roles in coupling motor proteins to mitochondria to mediate its transport. The uniform and opposite microtubule polarity in anterior vs. posterior dendrites in PVD and in the axon vs. dendrite in DA9 makes it possible to cleanly separate the functions of kinesin-1 from dynein-mediated transport. Based on both genetic and

biochemical data, our results are consistent with the notion that two distinct adaptor complexes exist for kinesin-1 and dynein.

For kinesin-1-mediated transport, MIRO-1, MTX-2, MTX-1, KLC-1, and KHC/UNC-116 are all required. First, both *mtx-2* and *mtx-1* are absolutely required for kinesin-1-mediated mitochondrial movement, as shown by the strong loss-of-function phenotypes in both PVD and DA9 neurons of these mutants. The posterior dendrite phenotypes of *mtx-2* and *mtx-1* are as strong as that of the *miro-1* mutants. Second, in DA9, metaxin mutations can enhance the *miro-1* null mutant phenotype, indicating metaxin's function is at least partially independent of MIRO-1 (Supplementary Fig. 2a–c). Third, metaxin mutants do not alter MIRO-1 expression level or its mitochondrial localization.

Similarly, the *miro-1* mutation does not change metaxin expression levels or their localization to mitochondria. Importantly, overexpression of MTX-1 or MTX-2 completely rescue the posterior dendritic mitochondrial phenotype in *miro-1* mutants, suggesting that the metaxin complex itself can be sufficient for the kinesin-1-mediated mitochondrial transport in the absence of *miro-1*. On the contrary, overexpression of *miro-1* does not rescue *mtx-1* or *mtx-2* mutant phenotypes, strongly suggesting that the metaxin complex is the essential component to link mitochondria to kinesin-1. Last, MTX-2, MTX-1, and KLC-1 can form a molecular complex in the absence of MIRO-1, suggesting that metaxins might be sufficient to link mitochondria to kinesin-1, mirroring the ability of metaxin overexpression to rescue the *miro-1* mutants. It should be noted that the *miro-1* mutant showed a near-complete loss of kinesin-mediated transport, indicating that it is indispensable in physiological conditions. Indeed, MIRO-1, MTX-2, MTX-1, and KLC-1 are found in the same molecular complex using purified proteins, confirming the existence of this super molecular adaptor complex. A recent study showed that Miro proteins are not required for TRAK/kinesin-dependent anterograde movement in mammalian cells, indicating that there should be Miro independent kinesin coupling mechanisms for mitochondria in the mammalian system[23].

Mammalian kinesin-1 is composed of two motor subunits (heavy chains, KHCs) and two light chains (KLCs)[34]. Usually, KLCs but not KHCs bind to cargo vesicles and proteins[34]. However, strong genetic evidence indicates that KHC can transport mitochondria in a light chain-independent manner[21], although kinesin light chains are found on mitochondria by several studies[38–40]. Indeed, KLC antagonizes KHC's ability to bind to Milton and is absent from the Milton-KHC complex[21]. *C. elegans* only contains a single Milton like molecule, TRAK-1, but the *trak-1* mutant does not show any defects in the mitochondrial distribution in PVD posterior dendrites or the DA9 axon (Fig. 5a–d and Supplementary Fig. 5f), which are both dependent on kinesin-1-mediated transport. Together, these data suggest that mitochondria might be transported by two different kinesin complexes, one of which contains Milton/TRAK but without KLC, the other contains KLC and both metaxins. The PVD and DA9 neurons rely on the latter complex.

For dynein-mediated transport, *miro-1*, *mtx-2*, *trak-1*, and *dli-1* are required. Similar to *Drosophila* and mammalian cells, the importance of MIRO-1 and TRAK-1 in dynein-mediated transport is supported by the strong loss of mitochondria from anterior PVD dendrites as well as the DA9 dendrite, both of which contain nearly pure minus-end-out microtubules. Interestingly, MTX-2 but not MTX-1 is also essential for dynein-mediated transport. We can detect a clear direct interaction between MIRO-1 and MTX-2 (Fig. 2k, l and Supplementary Fig. 3j), suggesting that these two mitochondrial proteins might exist in a complex. In contrast to the kinesin-1 adaptor complex, for dynein, MIRO-1's function is essential because overexpression of MTX-2 could not rescue the *miro-1* phenotype. Consistent with this result, Miro1 is required for TRAK2-dependent retrograde redistribution of mitochondria in mammalian neurons[23].

We further investigated if MTX-1 and MTX-2's function in mitochondria transport is conserved in vertebrate neurons. The fact that we have also detected mitochondrial trafficking defects in both anterograde and retrograde directions in human neurons suggests that MTX-1 and MTX-2 homologs are likely to play important roles in both kinesin-1- and dynein-mediated mitochondrial transport (Fig. 7a–e). Compared with the null mutant alleles in *C. elegans*, the RNAi approach likely represents a partial loss of MTX-1 and MTX-2. We have chosen the sparse transfection method in iPSC-derived neurons in order to track individual mitochondrial movement within defined axons and determining the directionality of movements[24,41], although we recognize that the CRISPR/Cas9 method potentially can generate more consistent knockdown and represents a more reliable way to assess protein function. Detailed analyses of MTX-1 and MTX-2 in vertebrate neurons will be needed to define the adaptor complexes responsible for anterograde and retrograde trafficking.

There are likely differences between the mammalian cells and worm neurons in terms of the exact motor-adaptor complexes responsible for mitochondria transport for several reasons. First, the mammalian TRAK proteins are required for both kinesin- and dynein-mediated transport. However, our data showed that *trak-1* is likely only required for dynein- but not kinesin-mediated mitochondria transport in *C. elegans* neurons. Second, the mechanism for localizing MTX-2 to mitochondria is likely different between worm neurons and mammalian cells. In worms, MTX-2 localizes to mitochondria independent of MTX-1 (Fig. 4b). On the contrary, one study showed that MTX-1 recruits MTX-2 to mitochondria in mammalian cell[32]. It is therefore plausible that knocking down MTX-1 in iPSC-derived neurons might indirectly affect MTX-2's localization to mitochondria, and thus affect dynein-mediated retrograde transport. Third, while *C. elegans* only have two metaxins, the human genome contains three metaxins.

The physiological significance of local mitochondria in neurites has been mostly studied in the axon. Mitochondria provide energy for efficient action potential firing, participating in synaptic transmission and regulating synaptic $Ca^{2+}$ levels[1,42]. However, the function of local mitochondria in dendrites has not been systematically studied. Mitochondrial trafficking mutants that reduce anterior PVD dendrites show a striking age-dependent degeneration. The phenotype is particularly strong in the distal half of the dendrite, suggesting that distal dendrites might suffer from depleted ATP due to the inefficient diffusion of ATP in neurites. Such intracellular energy gradients have been reported in mouse embryonic fibroblasts (MEFs), which are indeed shaped by mitochondrial distribution[43]. The exact mechanisms for such degeneration remain to be studied.

In summary, we have identified previously unknown mitochondrial adaptor proteins that play essential roles in connecting mitochondria to molecular motors. MTX-2 and MIRO-1 are core components of both kinesin-1 and dynein adaptor complexes, whereas MTX-1 and TRAK-1 specify the adaptors for kinesin-1 and dynein, respectively.

## Methods

**Worm genetics and DNA manipulations**. All the transgenes and mutants were from wild-type strain N2 Bristol. Worms were cultured on the nematode growth medium (NGM) plates seeded with *Escherichia coli* OP50 at 20 °C. Integrated *ser-2*P3 > *tomm-20(1-54aa)::gfp*; *ser-2*P3 > *myri-mcherry* transgenes, visualizing mitochondria distribution in PVD neuron, were mutagenized with 50 mM ethyl methane sulfonate (EMS). 15 mutants including *wy50250* and *wy50256* alleles were isolated from an F2 semi-clonal screen of 4000 haploid genomes. SNP mapping and transgene rescue experiments were performed following standard protocols[44,45].

CRISPR/Cas9-assisted knockout and knockin worms were generated[46]. The guide RNA target sequence was selected according to the design tool (http://crispor.tefor.net/). For GFP::MIRO-1 knockin animals, *gfp* was inserted between *miro-1* promoter and the coding sequence. *gfp* fragment was inserted between ~1.3 kb upstream and ~1.3 kb downstream homologous sequence amplified from the N2 genomic DNA, and then assembled into pPD95.77 backbone as *gfp* knockin repair template plasmid. Synonymous mutations were introduced to Cas9 target site to prohibit the cleavage of repair template by Cas9. The *gfp* knockin repair template plasmid (30 ng/μl), Cas9-gRNA plasmid (50 ng/μl), and selective marker P*ord-1* > *gfp* (30 ng/μl) were co-injected into N2 worms. The *gfp* knockin worms were identified and confirmed by PCR and Sanger sequencing.

Single-copy transgenes were generated using the miniMos-mediated single-copy insertion method[47]. *gfp* was fused in frame to *mtx-1* or *mtx-2* N-terminus, and then the *gfp::mtx-1*, *gfp::mtx-2*, and *miro-1* coding sequence were inserted into pWZ347 (a modified version of pCFJ909) with the *ser-2*P3 promoter (~1.6 kb) and the *unc-54* 3′UTR, respectively. The single-copy insertion plasmid pYS109, pYS120

or pYS436 (20 ng/μl), transposase plasmids pCFJ601 (50 ng/μl), and selective marker P*ord-1* > *gfp* (10 ng/μl) were co-injected into *unc-119(ed4)* worms. The uncoordinated phenotype of *unc-119(ed4)* worms was rescued in integration lines. Integration lines were further confirmed by PCR and Sanger sequencing. At least two independent lines were isolated for each construct and representative lines (*wySi50001[ser-2P3> gfp::mtx-2]*, *wySi50002[ser-2P3> gfp::mtx-1]*, and *wySi50004 [ser-2P3> miro-1]*) were selected for further characterization.

Transgenes for tissue-specific knockdown were constructed[48]. Both the forward and reverse complement *unc-116* cDNA were inserted into pPD95.77 driven by the ~1.6 kb *ser-2*P3 promoter. Two fragments (*ser-2*P3 > *unc-116* and *ser-2*P3 > *unc-116* reverse complement) were independently amplified, and purified by gel extraction. PVD-specific *unc-116* knockdown strain was generated by microinjecting the two gel extracted constructs (~100 ng/μl) mixed with selective marker P*myo-2* > *mcherry* (~1 ng/μl) into *wyIs50054* strain.

All of the strains and the plasmids used are listed in Supplementary Tables 1 and 2, respectively. Primers used for genotyping and plasmids construction are listed in Supplementary Tables 3 and 4, respectively.

**Mitochondrial isolation, immunoprecipitation, and western blotting.** Cell mitochondria isolation kit (Beyotime) was used for mitochondrial isolation from HEK293T cells. The experiment was carried out according to the instructions of the kit. In brief, the cultured cells were collected and mechanically homogenized. After centrifugation at $1000 \times g$ for 10 min at 4 °C, crude supernatant was spun at $3500 \times g$ for 10 min at 4 °C to pellet intact mitochondria. The acquired pellet was resuspended in lysis buffer (Beyotime) with 0.25 mM PMSF and protease inhibitors as the mitochondrial fraction. Then the supernatant was collected and centrifuged at $12,000 \times g$ for 10 min at 4 °C. The new supernatant was regarded as the cytosolic fraction. Both mitochondrial fraction and cytosolic fraction were incubated with 50 μl mouse anti-Miro1 antibody (ab188029, AbCam) at 4 °C overnight, respectively, and then were mixed with 200 μl washed Pierce Protein A/ G Magnetic Beads (88802, Thermo Fisher Scientific) for 2 h at 4 °C, respectively. Beads were then washed four times with wash buffer (150 mM NaCl, 50 mM Tris-HCl) with 0.25 mM PMSF and protease inhibitor cocktail. Then the beads were resuspended in SDS loading buffer and boiled for 10 min before the samples were loaded into SDS-PAGE gel. After electrophoresis, proteins were transferred onto a PVDF membrane. Transferred membranes were blocked for 1 h in Tris Buffered Saline Tween (TBST) containing 5% fat-free milk, and then incubated with the following primary antibodies: mouse anti-Miro1 (WH0055288M1, Sigma-Aldrich) at 1:1000, mouse anti-Metaxin1 (sc-514469, Santa Cruz Biotechnology) at 1:500, mouse anti-Metaxin2 (sc-514231, Santa Cruz Biotechnology) at 1:500, mouse anti-Tom20 at 1:500 (sc-17764, Santa Cruz Biotechnology), at room temperature for 4 h. After washing three times using TBST, 10 min each, the membranes were incubated with the HRP-conjugated Affinipure Goat Anti-Mouse IgG(H + L) (SA00001-1, Proteintech) at 1:2500 at room temperature for 2 h. After three times washing of TBST, pictures were taken.

**Co-immunoprecipitation assay.** Co-immunoprecipitation assay was used to detect the protein–protein interaction in vivo. The plasmids expressing Flag-tagged bait proteins and HA-tagged prey proteins were cloned into pNTAP vector individually. Each bait and prey pair plasmids were cotransfected into HEK293T cells and the transfected cells were harvested after 48 h. Then the cells were lysed using NP40 Cell Lysis Buffer (Thermo Scientific™ Catalog #FNN0021) with protease inhibitors (Sigma Catalog #P8340) on ice for 30 min. The lysates were pre-cleared with centrifugation at $12,000 \times g$ for 10 min at 4 °C. The supernatant of each sample was mixed with 50 μl ANTI-FLAG® M2 Affinity Gel (Merck Catalog #A2220). The mixture was incubated for 4 h at 4 °C with gentle rotation. Before adding to the mixture, the ANTI-FLAG® M2 Affinity Gel was washed with NP40 Cell Lysis Buffer three times. After rotation, the beads were washed by NP40 Cell Lysis Buffer with protease inhibitors four times. The bound proteins were eluted with SDS-PAGE buffer by incubating the sample for 10 min at 98 °C. The immunoprecipitates were subjected to Western blotting using the indicated antibodies.

**Protein expression and purification and analytical gel-filtration analysis.** DNA sequences encoding *C. elegans* MIRO-1 (residues 1–602), MTX-1 (residues 1–281), full-length MTX-2, and the TPR domain of KLC-1 (residues 214–502) were each cloned into a modified pET32a vector. For co-expression of MTX-1 and MTX-2, the corresponding genes were cloned into the pRSFDuet-1 vector. Recombinant proteins were expressed in *E. coli* BL21 (DE3) codon plus host cells at 16 °C. The His6-tagged or Trx-His6-tagged fusion proteins were purified by Ni$^{2+}$-Sepharose 6 Fast Flow (GE Healthcare) affinity chromatography followed by size-exclusion chromatography (Superdex-200 26/60, GE Healthcare). For analytical gel-filtration analysis, protein samples were loaded on the Superdex-200 10/300 column (GE Healthcare) in the buffer containing 25 mM HEPES pH 7.4, 100 mM NaCl, 1 mM DTT, and 1 mM EDTA.

**GST pull-down assay.** All proteins used in the GST pull-down assay were expressed in *E. coli*. The cell lysates containing GST or GST-tagged MIRO-1 proteins were mixed with cell lysates containing His-tagged or Trx-tagged proteins

(Trx-MTX-2, co-expressed Trx-MTX-1/Trx-MTX-2, co-expressed Trx-MTX-1/ Trx-MTX-2 with additional Trx-MTX-1, or co-expressed Trx-MTX-1/Trx-MTX-2 and His-KLC-1-TPR), respectively, in each experiment. The cell lysates containing co-expressed GST/Trx-MTX-2 or co-expressed GST-MTX-1/Trx-MTX-2 proteins were mixed with cell lysates containing His-tagged KLC-1-TPR proteins, respectively, in PBS with Glutathione Sepharose 4 Fast Flow (GE Healthcare) beads at 4 °C overnight. After washing the beads five times with PBS, the proteins captured by the beads were eluted with PBS containing 20 mM glutathione and detected by SDS-PAGE with Coomassie-blue staining.

**RNAi.** The following siRNA were used: Metaxin-1 (sc-88250, Santa Cruz Biotechnology); Metaxin-2(sc-95035, Santa Cruz Biotechnology); Trak1 (124828, Thermo Fisher Scientific); Trak2 (sc-60763, Santa Cruz Biotechnology); non-targeting control siRNA (SIC001, Sigma-Aldrich).

**Neuronal derivation from iPSCs and transfection.** The iPSC line in this study has been fully characterized by the previous studies[49,50]. iPSCs were derived to midbrain dopaminergic neurons[49,51–54]. In brief, neurons were generated using an adaptation of the dual-smad inhibition method with the use of smad inhibitors dorsormorphin (Sigma) and SB431542 (Tokris), and the addition of GSK3β inhibitor CHIR99021 (Stemgent). SHH was replaced with the smoothened agonist SAG. To gain a higher purity of neural precursor cells, 12 days after neural induction, rosette-forming neuroectodermal cells were manually lifted and detached en bloc, and then cultured in suspension in a low-attachment dish (430589, Corning Inc.) with N2 medium with 20 ng/ml BDNF, 200 μM Ascorbic Acid, 500 nM SAG, and 100 ng/ml FGF8a. On day 17, neurons were transferred onto poly-ornithine and laminin-coated glass coverslips in a 24-well plate. On day 18, medium was switched to N2 medium supplemented with 20 ng/ml BDNF, 200 μM Ascorbic Acid, 20 ng/ml GDNF, 1 ng/ml TGFβ3, and 500 μM Dibutyryl-cAMP for the maturation of dopaminergic neurons. Neurons were used at day 21–22 after neuronal induction, when about 80–90% of total cells expressed the neuronal marker TUJ-1, and 12% of total cells expressed TH and markers consistent with ventral midbrain neuronal subtypes[50,55].

For transfection, on day 18–19 after neuronal induction, the culture medium was replaced with Opti-MEM (Gibco) prior to transfection. 100 nM RNA, 0.5 μg EGFP, and 0.5 μg mito-dsRed, or 5 μl Lipofectamine 2000 was diluted in Opti-MEM at room temperature (22 °C) to a final volume of 50 μl in two separate tubes, and then contents of the two tubes were gently mixed, incubated for 20 min at room temperature, and subsequently added onto neurons. After transfection for 6 h, Opti-MEM containing RNA-DNA-Lipofectamine complexes was replaced with regular N2 medium. After transfection for 3 days, neurons were imaged.

**Live image acquisition and quantification**

*Worm.* *C. elegans* were anesthetized using 1 mg/ml levamisole in M9 buffer, and then mounted on 3% (w/v) agar pads. All images for control and the corresponding mutants were taken by the spinning disk confocal imaging system (Yokogawa CSU-X1 Spinning Disk Unit) at the same condition. For most images showed in this study, late L4 stage animals were imaged using a 100×/1.45 NA objective, and then processed by Image J. For GFP::MIRO-1 expression level quantification, gastrulation stage embryos were imaged, and fluorescence intensity of the single layer of the largest cell number was quantified by Image J. Mitochondrial dynamics were recorded around the cell body or primary dendrites of L3 or early L4 stage worms. For movies around the cell body region, dynamic images were recorded continually with 6-s interval for 30 min. Mitochondrial dynamics were classified into three events: "extension", "fission", and "transport". The frequency of events was calculated as the ratio between the worms displaying an event within 30 min of recording and the total number of worms recorded. In primary dendrite, time-lapse movies were taken with 3-s interval for 10 min. Mitochondrial behaviors were characterized by state (mobile or stable), the direction of motion (anterograde or retrograde), velocity. The velocity of mitochondrial movement was quantified by Imaris (Bitplane). For mitochondrial distribution quantification, worms at day 1 stage were examined under 63×/1.4 NA objective using an Axio Imager M2 microscope (Carl Zeiss).

*Human cell.* Neurons on glass coverslips were placed in a 35-mm petridish containing the Hibernate E low-fluorescence medium (BrainBits) on a heated stage of 37 °C, and imaged with a 63×/N.A.0.9 water-immersion objective with excitation at 561 nm or 488 nm. We imaged axons positive for both mito-dsRed and EGFP which likely took up RNA duplex. Axons longer than 50 μm were selected for recording. Time-lapse movies were obtained continually with 5-s intervals for 3–5 min. For quantification, kymographs were generated from time-lapse movies by ImageJ, representing a 100-s period. Each kymograph was then imported into a macro written in Labview (NI, TX), and individual mito-dsRed puncta were traced using a mouse-driven cursor at the center of the mito-dsRed object. Using Matlab (The MathWorks, MA), we determined the following parameters: (1) the instantaneous velocity of each mitochondrion, (2) the average velocity of those mitochondria that are in motion, (3) the percent of time each mitochondrion is in motion, (4) stop frequency, and (5) turn back frequency.

**Cell culture, transfection, and biochemistry**. HEK293T cells were cultured in high-glucose DMEM (Invitrogen) supplemented with 10% fetal bovine serum (900-108, Gemini Bio Products) and maintained in a 37 °C, 5% $CO_2$ incubator with the humidified atmosphere. HEK293T cells were transfected with 50 nM siRNA using calcium phosphate[49]. Cell lysates were collected after transfection for 3 days, and immunoblotted with the following antibodies: mouse anti-Metaxin-1 (sc-514469, Santa Cruz Biotechnology) at 1:500, mouse anti-Metaxin-2 (SC-514231, Santa Cruz Biotechnology) at 1:500, rabbit anti-Trak1 (HPA005853 Sigma-Aldrich) at 1:1000, rabbit anti-Trak2 (PA5-31459, Thermo Fisher) at 1:1000, mouse anti-ATP5β (AB14730, AbCam) at 1:3000, and mouse anti-β-actin (A00702, Genscript) at 1:5000.

**Statistics and reproducibility**. Statistical analyses were performed using Graph-Pad Prism 9.0.0.121. Two-tailed unpaired Student's $t$ test, one-way ANOVA with Tukey's multiple comparisons test, or Fisher's exact test was used to calculate the $p$ values. $*p < 0.05$, $**p < 0.01$, $***p < 0.001$, and $****p < 0.0001$ are considered significant. The types of the statistical tests, the sample size, exact $p$ values, and statistical significance are reported in the figures and corresponding figure legends. Data shown in graphs represent the mean ± standard error of the mean (SEM), and individual data points are plotted whenever possible. Statistical analysis is conducted on data from at least three biologically independent experimental replicates excepted when otherwise stated. Micrographs and biochemical images shown in the figures are representative of three or more independent experiments with similar results. The source data for statistical analyses can be found in the source data file.

**Reporting summary**. Further information on research design is available in the Nature Research Reporting Summary linked to this article.

## Data availability
All relevant data are available from the authors. Source data are provided with this paper.

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

## Acknowledgements

We are grateful to the *Caenorhabditis* Genetics Center for strains. We thank Yan Teng and Yun Feng from Center for Biological Imaging (CBI), Institute of Biophysics, for technical support with confocal imaging and image analysis. This work was supported by grants from the National Key R&D Program of China (2017YFA0102601 and 2017YFA0503501), the Strategic Priority Research Program of CAS (XDB37020302), and the National Natural Science Foundation of China (31571061, 31771138, and 31770786) to Xm. W and W.F.; as well as a grant from the Beijing Municipal Science & Technology Commission (Z181100001518001) and the National Natural Science Foundation of China (31829001) to K.S. K.S. is an investigator of the Howard Hughes Medical Institute.

## Author contributions

K.S., Xm.W. conceived the study. Y.Z. carried out most experiments. E.S., W.W., and C.H. contributed to some experiments. Y.Z., K.S., Xm.W., W.F., and Xn.W. designed and interpreted experiments. K.S. and Xm.W. wrote the manuscript with contributions from Y.Z., W.F., Xn.W., and E.S. All authors contributed to the discussion and analysis of the obtained results and the input to the final manuscript.

## Competing interests

The authors declare no competing interests.
