## [Peer Review File · Nature Communications]

Reviewers' comments:

Reviewer #1 (Remarks to the Author):

Mitochondria are transported along microtubules into dendrites and axon of neurons. They expose proteins on their surface that interact with motor proteins kinesin and dynein. Miro and its adaptor protein TRAKs are well-described components. Zhao and colleagues report here an unexpected function of metaxins, MTX-1 and MTX-2, as core subunits of adaptor complexes in the transport of mitochondria along the microtubules. Previously, it was described that metaxins are part of the SAM complex that inserts proteins into the outer membrane. The authors report that metaxins form the core of two distinct adaptor complexes that promote mitochondrial transport. MTX-2/MIRO-1 and TRAK-1 mediate dynein-dependent transport, while MTX-1/MTX-2, and MIRO-1 mediate kinesin-dependent transport of mitochondria. The authors showed that the metaxins are important for mitochondria trafficking in *C. elegans* and human cell lines. Loss of metaxins impairs transport of mitochondria in neuronal cells. The findings are novel and potentially highly interesting. The authors should add some additional data to substantiate their conclusions about the molecular mechanisms how metaxins promote transport of mitochondria.

1. The authors used mainly recombinantly expressed and purified proteins to analyze protein-protein in vitro. However, the authors should demonstrate that the detected interactions exist in cells and occur on the mitochondrial surface. For instance, is the binding of motor proteins/microtubule to mitochondria affected in the metaxin mutant cells? Do metaxins interact with Miro and adaptor proteins in mitochondria? Additional experimental data is required to tackle this issue.

2. The presented binding assays in HEK293T cells (e.g. MTX2-TRAK1, MTX2-MIRO1) are not entirely clear. The interactions are weak and require further controls to show the specificity (Extended Figures 2f and 4g). The authors should include negative controls in these assays that are not co-precipitated. In addition, the authors should perform reverse pulldowns.

3. Metaxins are known to interact with SAM50 to promote protein import into mitochondria. Whether the newly described role of metaxins in transport of mitochondria is connected to protein import remains unclear. Is SAM50 required for transport of mitochondria?

4. The authors should show that the steady state levels of MIRO-1 are not altered in MTX-1 and MTX-2 mutants to exclude indirect effects due to impaired protein biogenesis.

Reviewer #2 (Remarks to the Author):

The manuscript by Zhao et al. (NCOMMS-19-2937437-T) expanded on the existing molecular mechanisms of neuronal mitochondrial trafficking by revealing metaxin1/2 as new adaptors for recruiting kinesin and dynein motors. The authors elegantly applied worm PVD sensory neurons and DA9 neurons, where the microtubule polarity in anterior vs. posterior dendrites or the axon vs. dendrite is uniformly and oppositely distributed. These unique patterns allow the authors to separate kinesin-1 from dynein-mediated mitochondrial transport in an in vivo model system. By genetic interaction and gene rescue analyses, combined with biochemical approaches, the authors characterized dendritic and axonal mitochondrial distribution (density) in *C. elegans*. They reported that MTX-1/2 play essential, but distinct, roles in coupling motor proteins to mitochondria to mediate

mitochondrial transport. The authors further confirmed the conserved function of MTX-1/2 in mitochondrial transport along axons of human iPSC-derived neurons. Furthermore, they provide evidence that mitochondrial transport deficiency in *miro-1*, *trak-1*, or *mtx-2* mutant neurons leads to anterior dendritic degeneration. Based on these data, they provide a model for two distinct adaptor complexes for kinesin-1 and dynein-driven mitochondrial transport: while MTX-2 and MIRO-1 are core components of both kinesin-1 and dynein adaptor complexes, MTX-1 and TRAK-1 specify the adaptors for kinesin-1 and dynein-driven transport, respectively.

Overall, this is an interesting study. Identification of new motor adaptors for mitochondrial transport may help solve the recent puzzle as to why loss of *Drosophila* Miro cannot fully abolish mitochondrial distribution to distal neurites and why TRAK1/2 are still recruited to mitochondria in *Miro1/2* double-knockout cells. Thus, additional mitochondrial adaptors or receptors are likely present in neurons. The manuscript is logically presented and most of the *in vivo* data are solid. It represents a great deal of work, as evidenced by the comprehensive and quantitative analyses of data from *in vivo* studies. While the model seems promising, further evidence is needed to provide the necessary mechanistic insights to support the model. I have three main concerns and several specific comments; addressing these would strengthen their study and thus make an important contribution to the field.

Main Concerns:

(1) While *in vivo* imaging in PVD dendrites and genetic interaction and rescue data of mitochondrial distribution are of high quality, the biochemical analyses supporting a mechanistic role of MTX-1/2 in mitochondrial transport is rather premature (see my specific comments below).

(2) The current study does not provide detailed characterization of *in vivo* mitochondrial transport or motility in PVD sensory neurons. Instead, they characterized mitochondrial distribution or density, which are phenotypes that could also be affected by mitochondrial maintenance or degenerative factors following genetic manipulation. This assumption is supported by the robust changes in dendritic integrity in PVD neurons following mutation of *miro-1*, *mtx-2*, or *trak-1* (Fig. 6a-d).

(3) The authors showed that knockdown of MTX-1 or -2 significantly compromised mitochondrial transport in both directions in mammalian neurons (Fig 7), thus providing more practical tools to validate the models of the two separate adaptor complexes to accommodate kinesin and dynein-driven mitochondrial transport in neurons. In particular, the authors can further test their model that MTX1/2 serve as new adaptors on the mitochondrial outer membrane to recruit kinesin and dynein, and not rely only on mitochondrial distribution. In addition, they showed MTX-1 mainly binds to MTX-2 and Miro-1 and mediates kinesin-driven mitochondrial transport in posterior dendrites through interaction with KLC-1, which raises the question as to how knockdown of MTX-1 in iPSC neurons affects dynein-driven retrograde transport along axons (Fig7).

Specific comments:

(1) Biochemical approaches including co-IP, GST-pull down, and the analytical gel-filtration assay do not sufficiently support direct interactions between MTX-1/2 and Miro-1 or Trak1, as the authors propose. Mapping specific interaction domains and characterizing mitochondrial transport or distribution by expressing the binding loss-of-function mutant may support their adaptor complex models.

(2) Data from GST-pulldown assays (Figs 2 and 4) and the weak co-IP'ed complex (Fig S2f) with Flag-based overexpression have some limitations in reflecting dynamic protein adaptor interaction on the surface of mitochondria. Co-IP assays using endogenous proteins isolated from a mammalian neuron system, combined with super-resolution imaging analysis, are necessary to validate the proposed complex of MTX-2 and Miro-1 on the surface of neuronal mitochondria.

(3) Fig4i and FigS2g: It is premature to conclude that Miro-1 binds the MTX-1/2 complex, although both MTX-1 and -2 look to be in ~1:1 stoichiometric ratio. A separate complex of Miro1-MTX-1 or

Miro1-MTX-2 cannot not be excluded. A dose-independent stoichiometric binding assay by increasing one binding partner would be helpful to confirm the tri-protein complex model shown in Fig4n. This also applies to the model shown in Fig5k.

(4) I do not understand why MTX-2 can partially substitute for Miro-1's function when it is overexpressed. Mechanistic details about potential competitive versus cooperative interaction of MTX-2 and Miro-1 in binding Trak-1 may help to support the proposed adaptor complex models.

(5) Since both miro-1 and mtx-2 mutants lead to anterior dendritic degeneration, a control for general transport, such as lysosome or endosome distribution in PVD neurons, should be added in Fig1 or FigS1 to exclude general transport defects in these mutants due to dendritic degeneration.

(6) In Fig1e and Fig S1c-h, the authors characterized mitochondrial dynamics near soma-dendritic regions into three events: extension, fission, and transport. To exclude a possible effect of a fission event on mitochondrial transport, I would suggest adding a drp-1 mutant as a control.

(7) The authors showed that MTX-1 binds MTX-2 and Miro-1 and mediates kinesin-driven mitochondrial transport in posterior dendrites through an interaction with KLC-1. The authors should explain why knockdown of MTX-1 in iPSC neurons affects dynein-mediated retrograde transport along axons (Fig7).

(8) Based on their biochemical results, MTX-2, Miro-1, and Trak-1 can form a tri-protein complex. Given that both MTX-2 and Miro-1 are OMM proteins that target to mitochondria independent of each other, and overexpressed MTX-2 can substitute Miro-1's function in mitochondrial transport, I am wondering whether MTX-2 can interact with Trak-1 in a Miro-1-independent manner. If this is the case, it would be interesting to know whether MTX-2 and Miro-1 competitively or simultaneously bind to Trak-1. Addressing this issue will advance our knowledge as to how these adaptors and receptors work together in recruiting motors to drive mitochondrial transport.

Minor issues:

(1) Some of the panels of Figs and Supl Figs were not arranged logically and did not appear sequentially in the main text. For example, FigS3 was cited in the text before FigS2.

(2) Title of Fig2: "The MTX-2/MIRO-1 complex is important for mitochondrial trafficking". This title does not represent the data presented. Instead, the figure mainly showed the relative rescue effect or genetic interaction. There is no data showing the complex between MTX-2 and Miro-1 from this figure.

(3) In FigS3j, the input band of Flag-MTX-1 is hardly seen.

(4) IP experiments in FigS3 and FigS4 need a negative control.

Reviewer #3 (Remarks to the Author):

The authors describe a new partner for neurite transport in C elegans neurons - metaxins: MTX-1 and MTX-2 are shown to be involved in mitochondrial transport in this system.

The human proof of this concept was performed in 1 line of iPSC derived midbrain neurons. The authors perform RNAi knockdown of MTX1, 2, TRAK1, 2 and measure axonal transport which is significantly impaired in all 4 conditions. Was not clear to me, how the authors controlled the efficiency of the knockdown in human iPSC derived neurons (since in most cases RNAi is not working efficiently in iPSC derived systems). A WB proof of knock down in the respective iPS cell line is crucial to make the claim (currently shown in HEK cells). Moreover, detailed information, about how the control was treated is necessary (exclude RNAi transfection effect).

The paper, mostly focusing on a *C. elegans* system is hard to read, especially, since the systems are changing frequently between *C. elegans*, HEK and other cell types.

Reviewer #4 (Remarks to the Author):

This is a well-done study that convincingly supports the authors' model of how metaxins promote mitochondrial transport in *C. elegans* (and likely in mammalian cells). The only significant critique I can make of the manuscript is the absence of experiments assaying the RNAi knockdown levels in the iPSC neurons, as they report for the HEK293 cells. The statistical analyses and controls are appropriate, and the work does add to our understanding of mitochondrial transport, particularly the specific role of MTX-1 in kinesin-dependent mitochondrial transport.

Reviewers' comments:

Reviewer #1 (Remarks to the Author):

Mitochondria are transported along microtubules into dendrites and axon of neurons. They expose proteins on their surface that interact with motor proteins kinesin and dynamin. Miro and its adaptor protein TRAKs are well-described components. Zhao and colleagues report here an unexpected function of metaxins, MTX-1 and MTX-2, as core subunits of adaptor complexes in the transport of mitochondria along the microtubules. Previously, it was described that metaxins are part of the SAM complex that inserts proteins into the outer membrane. The authors report that metaxins form the core of two distinct adaptor complexes that promote mitochondrial transport. MTX-2/MIRO-1 and TRAK-1 mediate dynein-dependent transport, while MTX-1/MTX-2, and MIRO-1 mediate kinesin-dependent transport of mitochondria. The authors showed that the metaxins are important for mitochondria trafficking in *C. elegans* and human cell lines. Loss of metaxins impairs transport of mitochondria in neuronal cells. The findings are novel and potentially highly interesting. The authors should add some additional data to substantiate their conclusions about the molecular mechanisms how metaxins promote transport of mitochondria.

Response: We thank the reviewer for recognizing the potential significance of our work.

1. The authors used mainly recombinantly expressed and purified proteins to

analyze protein-protein in vitro. However, the authors should demonstrate that the detected interactions exist in cells and occur on the mitochondrial surface. For instance, is the binding of motor proteins/microtubule to mitochondria affected in the metaxin mutant cells? Do metaxins interact with Miro and adaptor proteins in mitochondria? Additional experimental data is required to tackle this issue.

Response: We thank the reviewer for this suggestion. We isolated mitochondria from HEK293T cells and performed pull-down experiments using a Miro1 antibody, and we indeed observed that Metaxin1 and Metaxin2 were present in the immunoprecipitate, but not another mitochondria outer membrane protein Tom20. We have included this new data in Extended Data Fig. 4.

We believe that in vivo imaging experiments are better suited to address the mitochondria-microtubule interaction. There are no precedents in reconstituting microtubule, mitochondria motors and related adaptor proteins using *C. elegans* material. Therefore, we did not attempt this experiment.

2. The presented binding assays in HEK293T cells (e.g. MTX2-TRAK1, MTX2-MIRO1) are not entirely clear. The interactions are weak and require further controls to show the specificity (Extended Figures 2f and 4g). The authors should include negative controls in these assays that are not co-precipitated. In addition, the authors should perform reverse pull-downs.

Response: We have included two negative controls: an empty vector control shown in Extended Data Fig. 3d and a mitochondrial outer membrane protein DRP-1 as a

control shown in Extended Data Fig. 3e.

We have also performed reverse pull-down experiments as the reviewer suggested.

The results confirm interactions between MTX-1, MTX-2, TRAK-1, and MIRO-1. The data are now presented in Extended Data Fig. 3, 4, and 5.

We would like to emphasize that these pull-down experiments are complementary to the *in vitro* reconstitution experiments using purified proteins. In the reconstitution experiment, we demonstrated direct binding between MTX-1, MTX-2, MIRO-1, and KLC-1 proteins.

3. Metaxins are known to interact with SAM50 to promote protein import into mitochondria. Whether the newly described role of metaxins in transport of mitochondria is connected to protein import remains unclear. Is SAM50 required for transport of mitochondria?

Response: We showed that the expression level and mitochondrial localization of MIRO-1 was normal in *mtx-1* and *mtx-2* mutants (Extended Data Fig. 3b, c), suggesting that metaxins' requirement in mitochondria transport is not through helping MIRO-1's localization to mitochondria, which is consistent with previous study that *gem1*, the ortholog of *miro-1* in *S. cerevisiae*, is independent with SAM complex for integration into the mitochondria outer membrane. ¹

We did follow the reviewer's suggestion to examine the only ortholog of SAM50 in *C. elegans*, GOP-3. In *gop-3* mutant, GFP::MTX-2 was completely absent from mitochondria and not detected anywhere in neurons, while the localization of

TOMM-20 was normal. Additionally, mitochondria were completely absent from neurites and stuck in the soma (see the following picture), similar to the *mtx-2* mutants. These new data suggest that SAM50 is required for the localization of MTX-2 on mitochondria.

scale bar: 10 μ m.

4. The authors should show that the steady state levels of MIRO-1 are not altered in MTX-1 and MTX-2 mutants to exclude indirect effects due to impaired protein biogenesis.

Response: To quantify the level and localization of MIRO-1 in vivo, we used CRISPR to knock GFP into the endogenous locus of *miro-1*, which resulted in punctate fluorescence signals in all cells colocalizing with TOMM-20::mCherry (Extended Data Fig. 3a). We compared the GFP::MIRO-1 fluorescence in wild type to that of the *mtx-1* and *mtx-2* mutants and found no difference between the wild type and mutants (Extended Data Fig. 3a-c). These results argue that the expression level and mitochondria localization of MIRO-1 is independent of MTX-1 and MTX-2.

Reviewer #2 (Remarks to the Author):

The manuscript by Zhao et al. (NCOMMS-19-2937437-T) expanded on the existing molecular mechanisms of neuronal mitochondrial trafficking by revealing metaxin1/2 as new adaptors for recruiting kinesin and dynein motors. The authors elegantly applied worm PVD sensory neurons and DA9 neurons, where the microtubule polarity in anterior vs. posterior dendrites or the axon vs. dendrite is uniformly and oppositely distributed. These unique patterns allow the authors to separate kinesin-1 from dynein-mediated mitochondrial transport in an in vivo model system. By genetic interaction and gene rescue analyses, combined with biochemical approaches, the authors characterized dendritic and axonal mitochondrial distribution (density) in *C. elegans*. They reported that MTX-1/2 play essential, but distinct, roles in coupling motor proteins to mitochondria to mediate mitochondrial transport. The authors further confirmed the conserved function of MTX-1/2 in mitochondrial transport along axons of human iPSC-derived neurons. Furthermore, they provide evidence that mitochondrial transport deficiency in *miro-1*, *trak-1*, or *mtx-2* mutant neurons leads to anterior dendritic degeneration. Based on these data, they provide a model for two distinct adaptor complexes for kinesin-1 and dynein-driven mitochondrial transport: while MTX-2 and MIRO-1 are core components of both kinesin-1 and dynein adaptor complexes, MTX-1 and TRAK-1 specify the adaptors for kinesin-1 and dynein-driven transport, respectively.

Overall, this is an interesting study. Identification of new motor adaptors for mitochondrial transport may help solve the recent puzzle as to why loss of

Drosophila Miro cannot fully abolish mitochondrial distribution to distal neurites and why TRAK1/2 are still recruited to mitochondria in Miro1/2 double-knockout cells. Thus, additional mitochondrial adaptors or receptors are likely present in neurons. The manuscript is logically presented and most of the in vivo data are solid. It represents a great deal of work, as evidenced by the comprehensive and quantitative analyses of data from in vivo studies. While the model seems promising, further evidence is needed to provide the necessary mechanistic insights to support the model. I have three main concerns and several specific comments; addressing these would strengthen their study and thus make an important contribution to the field.

Response: We thank the reviewer for the positive comments.

Main Concerns:

(1) While in vivo imaging in PVD dendrites and genetic interaction and rescue data of mitochondrial distribution are of high quality, the biochemical analyses supporting a mechanistic role of Miro-1/2 in mitochondrial transport is rather premature (see my specific comments below).

Response: Please see our response to specific comments.

(2) The current study does not provide detailed characterization of in vivo mitochondrial transport or motility in PVD sensory neurons. Instead, they characterized mitochondrial distribution or density, which are phenotypes that

could also be affected by mitochondrial maintenance or degenerative factors following genetic manipulation. This assumption is supported by the robust changes in dendritic integrity in PVD neurons following mutation of *miro-1*, *mtx-2*, or *trak-1* (Fig. 6a-d).

Response: We thank the reviewer for this question. First of all, we have examined the defects of mitochondria localization in the *mtx-1* and *mtx-2* mutants in multiple time points during the development and maintenance. The phenotypes of the mutants are very strong from early developmental stage (L3 and L4 larval stage during the outgrowth of neurites) to young adult and mature adult stage (maintenance stage). In other words, there is not any stage where significant amount of mitochondria can be detected in corresponding neurites in the *mtx-2* and *mtx-1* mutants. In stark contrast, the degeneration of neurites only occurs in mature adult animals, several days after we detect the mitochondria localization defects. Therefore, it is extremely unlikely that the neurite degeneration phenotypes cause the mitochondria transport defect.

We have also attempted to perform time-lapse imaging to directly measure the transport of mitochondria in wild type and *mtx-2* mutants. We were able to observe bidirectional mitochondria transport events in both anterior and posterior PVD dendrites in wild type (Extended Data Fig. 1). The mitochondria movements are rare events that only occur infrequently. We recorded a total of more than one hundred hours of movies and observed about 70 movement events. However, we could not record any movement mitochondria event in *mtx-2* mutant despite of dozens of

hours of recording, which precludes us from characterizing the transport parameters (Extended Data Fig. 1). These results further argue that the transport deficit of *mtx-2* mutants is extremely strong.

(3) The authors showed that knockdown of MTX-1 or -2 significantly compromised mitochondrial transport in both directions in mammalian neurons (Fig 7), thus providing more practical tools to validate the models of the two separate adaptor complexes to accommodate kinesin and dynein-driven mitochondrial transport in neurons. In particular, the authors can further test their model that MTX1/2 serve as new adaptors on the mitochondrial outer membrane to recruit kinesin and dynein, and not rely only on mitochondrial distribution. In addition, they showed MTX-1 mainly binds to MTX-2 and Miro-1 and mediates kinesin-driven mitochondrial transport in posterior dendrites through interaction with KLC-1, which raises the question as to how knockdown of MTX-1 in iPSC neurons affects dynein-driven retrograde transport along axons (Fig7).

Response: In mammalian cell, there are multiple metaxins in the genome. The experiments proposed by this reviewer are interesting and definitely worth doing for a future study but it is not our intention to perform a thorough analysis of the vertebrate metaxins. There are three metaxins in human. We would need to characterize the expression, establish stringent loss-of-function reagents and understand how they correspond to the *C. elegans* MTX-1 and MTX-2.

This manuscript is the first study of metaxins' function in mitochondria transport.

We have thoroughly characterized the genetics and biochemical requirement of MTX-1 and MTX-2 in *C. elegans*. The human neuron experiments were meant to be an initial test of the involvement of the homologs of MTX-1 and MTX-2 in mitochondria transport. We would like to suggest that a thorough characterization of the metaxins in human cells is outside of the scope of this study.

Specific comments:

(1) Biochemical approaches including co-IP, GST-pull down, and the analytical gel-filtration assay do not sufficiently support direct interactions between MTX-1/2 and Miro-1 or Trak1, as the authors propose. Mapping specific interaction domains and characterizing mitochondrial transport or distribution by expressing the binding loss-of-function mutant may support their adaptor complex models.

Response: We agree that these domain dissection experiments will help us to further understand how the proteins interact with each other. However, we respectfully disagree that these experiments will be necessary for the publication of this manuscript. MTX-1 and MTX-2 are fairly small proteins without any predicted domains. The proposed experiments require structural information to ensure that any interaction-perturbing mutations will not affect the folding of proteins. We would like to suggest that those are for future studies.

(2) Data from GST-pulldown assays (Figs 2 and 4) and the weak co-IP' ed complex (Fig S2f) with Flag-based overexpression have some limitations in reflecting

dynamic protein adaptor interaction on the surface of mitochondria. Co-IP assays using endogenous proteins isolated from a mammalian neuron system, combined with super-resolution imaging analysis, are necessary to validate the proposed complex of MTX-2 and Miro-1 on the surface of neuronal mitochondria.

Response: Following this suggestion, we performed a pull-down experiment using the MIRO1 antibody and mitochondria fraction from HEK293T cells. We detected Metaxin1 and Metaxin2 but not another mitochondria outer membrane protein Tom20 (Extended Data Fig. 4i). This result further strengthens the existing pull-down experiments and suggests that MIRO-1 interacts with MTX-1/2 on the mitochondria surface.

(3) Fig4i and FigS2g: It is premature to conclude that Miro-1 binds the MTX-1/2 complex, although both MTX-1 and -2 look to be in ~1:1 stoichiometric ratio. A separate complex of Miro1-MTX-1 or Miro1-MTX-2 cannot not be excluded. A dose-independent stoichiometric binding assay by increasing one binding partner would be helpful to confirm the tri-protein complex model shown in Fig4n. This also applies to the model shown in Fig5k.

Response: We thank the reviewer for this suggestion. We have performed the additional experiment as suggested by changing the MTX-1 dosage and found no visible difference in the amount of MTX-1 in the complex, confirming that these three proteins indeed exist in the same complex (Extended Data Fig. 4j).

(4) I do not understand why MTX-2 can partially substitute for Miro-1' s function when it is overexpressed. Mechanistic details about potential competitive versus cooperative interaction of MTX-2 and Miro-1 in binding Trak-1 may help to support the proposed adaptor complex models.

Response: We agree with the reviewer that these data suggest that there might be cooperative effect between MTX-2 and MIRO-1 in binding to TRAK-1. In our opinion, it is very unlikely that MTX-2 and MIRO-1 has competitive relationship because loss of either protein generated the same phenotype-loss of mitochondria transport. In addition, they form a protein complex. We agree that testing potential cooperative binding using purified proteins should provide additional mechanistic insights of this complex. However, we were not able to purified recombinant TRAK-1. To the best of our knowledge, no lab has been able to produce purified full-length, functional TRAK proteins in any organisms.

(5) Since both *miro-1* and *mtx-2* mutants lead to anterior dendritic degeneration, a control for general transport, such as lysosome or endosome distribution in PVD neurons, should be added in Fig1 or FigS1 to exclude general transport defects in these mutants due to dendritic degeneration.

Response: We thank the reviewer for this suggestion. We examined the localization of early endosome (GFP::RAB-5) and late endosome (GFP::RAB-7) in *miro-1* and *mtx-2* mutants. GFP::RAB-5 and GFP::RAB-7 puncta are distributed along the anterior and posterior dendrite with concentration in the proximal neurites. Both

miro-1 and *mtx-2* mutants showed distribution patterns of these two markers that are indistinguishable from those of control, further confirming that their effects on mitochondria localization are specific. The new data are added to Extended Data Fig. 1.

(6) In Fig1e and Fig S1c-h, the authors characterized mitochondrial dynamics near soma-dendritic regions into three events: extension, fission, and transport. To exclude a possible effect of a fission event on mitochondrial transport, I would suggest adding a *drp-1* mutant as a control.

Response: We thank the reviewer for this suggestion. We examined the phenotype of *drp-1* mutant, which showed a very interesting phenotype that was clearly distinct from those of *mtx-1*, *mtx-2*, and *miro-1* mutants. In *drp-1* mutant PVD, mitochondria form a long tubule that extended from the soma to the neurites (see the following picture). In wild type animals we observed that mitochondria extended short tubules into dendrite, followed by successive fission events to release mitochondria puncta into the neurites (Extended Data movie 1). These short tubules are absent in the *mtx-2* and *miro-1* mutants (Extended Data Fig. 1i, j and movie 2), suggesting that the extension of mitochondria tubules relies on motor coupling. It is highly likely that the long mitochondria tubule in the *drp-1* mutant is caused by motor activity pulling the mitochondria but failure to undergo fission. Because of the distinct loss-of-function phenotypes, the *mtx-2* and *miro-1* mutant phenotype is not caused by fission defects.

scale bar: 10 μm .

(7) The authors showed that MTX-1 binds MTX-2 and Miro-1 and mediates kinesin-driven mitochondrial transport in posterior dendrites through an interaction with KLC-1. The authors should explain why knockdown of MTX-1 in iPSC neurons affects dynein-mediated retrograde transport along axons (Fig7).

Response: There are likely differences between the mammalian cells and worm neurons in terms of the exact motor-adaptor complexes responsible for mitochondria transport for several reasons. First, the mammalian TRAK proteins are required for both kinesin- and dynein-mediated transport. However, our data clearly showed that *trak-1* is likely only required for dynein- but not kinesin-mediated mitochondria transport in *C. elegans* neurons. Second, the mechanism for localizing MTX-2 to mitochondria is likely different between worm neurons and mammalian cells. In worms, MTX-2 localizes to mitochondria independent of MTX-1 (Fig. 4b). On the contrary, one study showed that MTX-1 recruits MTX-2 to mitochondria in mammalian cell². It is therefore plausible that knocking down MTX-1 in iPSC-derived neurons might indirectly affect MTX-2's localization to mitochondria, and thus affect dynein-mediated retrograde transport. Third, while *C. elegans* only have two metaxins, human genome contains three

metaxins.

(8) Based on their biochemical results, MTX-2, Miro-1, and Trak-1 can form a tri-protein complex. Given that both MTX-2 and Miro-1 are OMM proteins that target to mitochondria independent of each other, and overexpressed MTX-2 can substitute Miro-1's function in mitochondrial transport, I am wondering whether MTX-2 can interact with Trak-1 in a Miro-1-independent manner. If this is the case, it would be interesting to know whether MTX-2 and Miro-1 competitively or simultaneously bind to Trak-1. Addressing this issue will advance our knowledge as to how these adaptors and receptors work together in recruiting motors to drive mitochondrial transport.

Response: This is an interesting idea. But our data do not support the notion that MTX-2 might interact with TRAK-1 in a MIRO-1-independent manner. First, overexpressed MTX-2 could not rescue dynein-mediated trafficking of mitochondria in the *miro-1* mutants, while the kinesin-mediated transport was rescued (Fig. 2h, i, and j). Second, loss of MTX-2 and MIRO-1 caused very similar, strong loss of transport for both kinesin- and dynein-mediated transport. It is very hard to imagine that they are competing with each other. The most parsimonious explanation is that MTX-2 and MIRO-1 are both required for the TRAK-1 mediated coupling to dynein.

Minor issues:

(1) Some of the panels of Figs and Supl Figs were not arranged logically and did not appear sequentially in the main text. For example, FigS3 was cited in the text before FigS2.

Response: We have changed them.

(2) Title of Fig2: "The MTX-2/MIRO-1 complex is important for mitochondrial trafficking" . This title does not represent the data presented. Instead, the figure mainly showed the relative rescue effect or genetic interaction. There is no data showing the complex between MTX-2 and Miro-1 from this figure.

Response: We have changed it to "MTX-2 interacts with MIRO-1 and functions in mitochondrial trafficking" .

(3) In FigS3j, the input band of Flag-MTX-1 is hardly seen.

Response: We have included a new gel in the new Extended Data Fig. 4a.

(4) IP experiments in FigS3 and FigS4 need a negative control.

Response: We have included two negative controls: an empty vector control shown in Extended Data Fig. 3d and a mitochondrial outer membrane protein DRP-1 as a control shown in Extended Data Fig. 3e.

Reviewer #3 (Remarks to the Author):

The authors describe a new partner for neurite transport in C elegans neurons - metaxins: MTX-1 and MTX-2 are shown to be involved in mitochondrial transport in this system.

The human proof of this concept was performed in 1 line of iPSC derived midbrain neurons. The authors perform RNAi knockdown of MTX1, 2, TRAK1, 2 and measure axonal transport which is significantly impaired in all 4 conditions. Was not clear to me, how the authors controlled the efficiency of the knockdown in human iPSC derived neurons (since in most cases RNAi is not working efficiently in iPSC derived systems). A WB proof of knock down in the respective iPS cell line is crucial to make the claim (currently shown in HEK cells). Moreover, detailed information, about how the control was treated is necessary (exclude RNAi transfection effect).

The paper, mostly focusing on a C. elegans system is hard to read, especially, since the systems are changing frequently between C elegans, HEK and other cell types.

Response: We thank the reviewer for the encouraging comments. We have confirmed the knockdown efficiency with control experiments. This new data are included as new Extended Data Fig. 6. We have also attempted to streamline the manuscripts to make it easier to read.

Reviewer #4 (Remarks to the Author):

This is a well-done study that convincingly supports the authors' model of how metaxins promote mitochondrial transport in C. elegans (and likely in mammalian

cells). The only significant critique I can make of the manuscript is the absence of experiments assaying the RNAi knockdown levels in the iPSC neurons, as they report for the HEK293 cells. The statistical analyses and controls are appropriate, and the work does add to our understanding of mitochondrial transport, particularly the specific role of MTX-1 in kinesin-dependent mitochondrial transport.

Response: We thank the reviewer for the positive comments. We have confirmed the knockdown efficiency with control experiments. This new data are included as new Extended Data Fig. 6.

1. Stojanovski, D., Guiard, B., Kozjak-Pavlovic, V., Pfanner, N. & Meisinger, C. Alternative function for the mitochondrial SAM complex in biogenesis of alpha-helical TOM proteins. *The Journal of cell biology* **179**, 881-893 (2007).
2. Armstrong, L.C., Saenz, A.J. & Bornstein, P. Metaxin 1 interacts with metaxin 2, a novel related protein associated with the mammalian mitochondrial outer membrane. *Journal of cellular biochemistry* **74**, 11-22 (1999).

REVIEWER COMMENTS

Reviewer #1 (Remarks to the Author):

The authors addressed some of my concerns in the revision and the manuscript improved. However, some points remain to be clarified.

The experiment presented in Figure 4i indicates that the interaction of endogenous MIRO with MTX1 and MTX2 could exist in cells. Since this is the only pulldown experiments with endogenous and non-tagged protein and crucial for the manuscript it should be presented in the main figures. The authors have to indicate in the figure legend how much load and elution are loaded. This information is important to assess whether the binding is efficient or not. Could the authors discuss how large is the fraction of MTX1 and MTX2 that binds to MIRO and to SAM50?

It remains unclear whether the two proposed adaptor complexes exist in mitochondria. This conclusion is based on in vitro binding studies with isolated proteins. Some additional data should support their conclusion.

The authors should show the steady state levels of MIRO in MTX1 and MTX2 mutant cells to exclude the possibility that impaired import of MIRO precursor via MTX1 or MTX2 leads to reduced MIRO levels. This question is not clearly addressed in the Extended Data Figure 3b.

Reviewer #2 (Remarks to the Author):

This revision (NCOMMS-19-2937437-T) has included additional experimental data. Although not all of my concerns have been well addressed, particularly those regarding mechanistic insights into the role of MTX1/2 in assembling two distinct motor-adaptor complexes for driving bi-directional mitochondrial transport (my main concerns #3, specific comments # 1, 4, and 7), I can accept the authors' argument that (1) mitochondrial movements are rare events in worm neurites, (2) it is technically challenging to pursue mechanistic studies in the worm model, and (3) there are likely differences between mammalian and worm neurons in terms of the motor-adaptor complexes. I agree with the authors that the current study will facilitate future mechanistic investigations in human or mouse models. I suggest that the authors discuss these limitations in the text.

I also offer two minor suggestions:

Extended Data Fig. 1m, n- it would be nice to provide quantitative data of RAB-5 and RAB-7 organelles along axons, rather than just showing images, thus strengthening their conclusion for a specific role of MTX1/2 in mitochondrial distribution in distal neurites.

Extended Data Fig. 4j- increased input ratio of Trx-MTX-1 over GST-MIRO-1 and Trx-MTX-2 should be clearly labeled. While this data may exclude the possibility that MTX-1/2 bind to Miro-1 in a competitive manner, there is no solid evidence supporting that Miro-1 directly interacts with both MTX-1 and MTX-2 simultaneously. This is rather a conceptual issue in the motor-adaptor field. It would be more cautious to rephrase "MIRO-1/MTX-1/2 complex links KHC/UNC-116 through light chain KLC-1" in lines 332-333.

Overall, I recommend for publication of the current revision in Nat Comms after the authors make a minor revision as I suggest.

Reviewer #3 (Remarks to the Author):

My previous concern was a lack of a Western Blot proof of knock down in the respective iPS cell line derived neurons. I still believe that this is crucial to make the claim (currently shown in HEK cells). It was not done within this revision.

The authors now added immunofluorescence images (Extension Fig. 6C). Immunofluorescence is not a linear method to prove the efficiency of knock-down and the level of knock down.

The paper uses 1 iPSC line and a rarely used knock-down technique within this field. The current standard in the field is to show several iPSC lines and use CRISPR-Cas based genome editing as a human stem cell based model.

Reviewer #4 (Remarks to the Author):

The authors have addressed my only significant concern.

REVIEWER COMMENTS

Reviewer #1 (Remarks to the Author):

The authors addressed some of my concerns in the revision and the manuscript improved. However, some points remain to be clarified.

The experiment presented in Figure 4i indicates that the interaction of endogenous MIRO with MTX1 and MTX2 could exist in cells. Since this is the only pulldown experiments with endogenous and non-tagged protein and crucial for the manuscript it should be presented in the main figures. The authors have to indicate in the figure legend how much load and elution are loaded. This information is important to assess whether the binding is efficient or not. Could the authors discuss how large is the fraction of MTX1 and MTX2 that binds to MIRO and to SAM50?

It remains unclear whether the two proposed adaptor complexes exist in mitochondria. This conclusion is based on *in vitro* binding studies with isolated proteins. Some additional data should support their conclusion.

Response: We thank the reviewer of this suggestion and we have moved the IP data to the new main figure Fig. 4h. And we have indicated the quantity of the loading of each lane in the figure legend. The fraction that binds to MIRO1 is about 1%, and 0.5% of the total protein for MTX1 and MTX2, respectively. In the pulldown experiments, detergent has to be used to solubilize mitochondria proteins. It is unclear how well this complex can sustain the extraction process. We did not do the pull down experiment of SAM50.

We believe there are substantial evidence to support the model that MTX-1, MTX-2, and MIRO-1 function on mitochondria to mediate its transport on microtubule. First, both MTX1 and MTX2 are well established mitochondrial proteins from the mammalian literature (Armstrong et al., 1999). We showed in this manuscript that both of them are unambiguously localized on mitochondria in neurons (Fig. 2a, 2b, and 4a). Second, the loss-of-function mutants of *mtx-1*, *mtx-2*, and *miro-1* showed dramatic, specific and similar phenotypes in mitochondria distribution in neurites. MTX-2 and MIRO-1 are both essential for mitochondria localization in axon and dendrites (Fig. 1b and 1g). The loss-of-function analyses and genetic interaction experiments strongly argue that these three genes function in the same genetic pathway. Third, we used purified proteins to show that MTX-1, MTX-2, and MIRO-1 form a complex. In our opinion, the reconstitution experiments with purified proteins is more stringent demonstration of direct binding between these proteins compared to *in vivo* pull down experiments. The reconstitution experiments also showed that MIRO-1, MTX-1, and MTX-2 exist roughly in stoichiometry ratios within the same complex. With the pull down experiments, there are other mitochondria proteins, which is not a demonstration of direct binding.

In summary, combining the *in vivo* localization of MTX-1, MTX-2, and MIRO-1 on mitochondria, the reconstitution of the protein complex *in vitro* with purified proteins and their very similar loss of function phenotypes, we have built a strong case that these proteins function together to transport mitochondria.

The authors should show the steady state levels of MIRO in MTX1 and MTX2 mutant cells to exclude the possibility that impaired import of MIRO precursor via MTX1 or MTX2 leads to reduced MIRO levels. This question is not clearly addressed in the Extended Data Figure 3b.

Response: To understand whether mitochondrial localization of MIRO-1 is dependent on MTX-1 or MTX-2, we made a GFP knockin strain in which GFP was inserted into the N-terminus of the endogenous *miro-1* gene. The GFP fluorescence in this strain localizes to mitochondria, demonstrated by both the stereotyped mitochondria pattern and its colocalization with TOMM-20::mCherry (Extended Data Figure 3a). Since this is a knockin strain, the fluorescence intensity accurately reflects both the level and the subcellular localization of MIRO-1. Measurements of GFP::MIRO-1 fluorescence intensity and localization pattern in the mutants showed no difference between wild-type and the *mtx-1* or *mtx-2* mutants. These data demonstrate that the mitochondrial localization of MIRO-1 and **the steady state level of MIRO-1 are not affected** in *mtx-1* and *mtx-2* mutants (Extended Data Figure 3c).

We have decided to use this method to examine the level of MIRO-1 instead of the traditional western blot analysis for several reasons. First, quantifying the GFP fluorescence in live animals avoids the multistep western blot analysis which includes homogenization, extraction, gel transfer, blotting and exposure. The GFP fluorescence is a quantitative and direct measure of protein levels in its *in situ* setting. Second, the GFP methods preserves the subcellular localization of MIRO-1 on mitochondria in different tissues. Therefore, when we are comparing the GFP level in wildtype against the mutants, we know that we are comparing the MIRO-1 of the same cell types and on mitochondria. With the western blot analyses, we would only be able to measure the overall level of MIRO-1 in the entire animal. Because of these considerations, we believe that we have shown that MTX-1 and MTX-2 do not control the level or localization of MIRO-1 on mitochondria with these fluorescence measurements.

Reviewer #2 (Remarks to the Author):

This revision (NCOMMS-19-2937437-T) has included additional experimental data. Although not all of my concerns have been well addressed, particularly those regarding mechanistic insights into the role of MTX1/2 in assembling two distinct motor-adaptor complexes for driving bi-directional mitochondrial transport (my main concerns #3, specific comments # 1, 4, and 7), I can accept the authors' argument that (1) mitochondrial movements are rare events in worm neurites, (2) it is technically challenging to pursue mechanistic studies in the worm model, and (3) there are likely differences between mammalian and worm neurons in terms of the motor-adaptor complexes. I agree with the authors that the current study will facilitate future mechanistic investigations in human or mouse models. I suggest that the authors discuss these limitations in the text.

Response: We have discussed this in the text.

I also offer two minor suggestions:

Extended Data Fig. 1m, n- it would be nice to provide quantitative data of RAB-5 and RAB-7 organelles along axons, rather than just showing images, thus strengthening their conclusion for a specific role of MTX1/2 in mitochondrial distribution in distal neurites.

Response: We have quantified the number of RAB-5 and RAB-7 puncta in PVD dendrites and found no differences between the wildtype and *mtx-1* and *mtx-2* mutants. We have included these data in the new Extended Data Figure 1n and 1p.

Extended Data Fig. 4j- increased input ratio of Trx-MTX-1 over GST-MIRO-1 and Trx-MTX-2 should be clearly labeled. While this data may exclude the possibility that MTX-1/2 bind to Miro-1 in a competitive manner, there is no solid evidence supporting that Miro-1 directly interacts with both MTX-1 and MTX-2 simultaneously. This is rather a conceptual issue in the motor-adaptor field. It would be more cautious to rephrase “MIRO-1/MTX-1/2 complex links KHC/UNC-116 through light chain KLC-1” in lines 332-333.

Response: We agree with this reviewer that our data exclude the possibility that MTX-1/2 bind to MIRO-1 in a competitive manner. MTX-1 and MTX-2 bind to each other strongly. We can envision that MTX-1 and MTX-2 both bind to MIRO-1 separately. Alternatively, probably more likely, MTX-2 directly binds to both MTX-1 and MIRO-1. We do not see why MTX-1 needs to bind directly to MIRO-1. But we do appreciate this reviewer pushing us to understand more detailed structure-function of these proteins. We are pursuing the structural work on the MTX-1, MTX-2, and MIRO-1. Once we have the structure of this complex, we will be in a much better position to perform structure-function analyses.

Overall, I recommend for publication of the current revision in Nat Comms after the authors make a minor revision as I suggest.

Reviewer #3 (Remarks to the Author):

My previous concern was a lack of a Western Blot proof of knock down in the respective iPSC cell line derived neurons. I still believe that this is crucial to make the claim (currently shown in HEK cells). It was not done within this revision.

The authors now added immunofluorescence images (Extension Fig. 6C). Immunofluorescence is not a linear method to prove the efficiency of knock-down and the level of knock down. The paper uses 1 iPSC line and a rarely used knock-down technique within this field. The current standard in the field is to show several iPSC lines and use CRISPR-Cas based genome editing as a human stem cell based model.

Response: The transfection method in iPSC-derived neurons (details in the Method section) was aimed to achieve a very low transfection efficiency (~1%). Sparse transfection of neuronal cultures is necessary to track individual mitochondrial movement. The axon of a single transfected neuron can be distinguished from those of neighboring neurons, and thus the

orientation of movement (anterograde or retrograde) can be determined unambiguously. In densely transfected cultures, the bundling of neurites makes it impossible to trace each axon back to the corresponding cell body to determine the direction of mitochondria movement (Wang and Schwarz, 2009 a, b). Therefore, our transfection method is ideal for live imaging mitochondrial transport, but not for western blotting which requires a high transfection efficiency. We performed immunostaining to validate RNAi efficiency on transfected neurons. We have used this method extensively to manipulate genes in the culture system (Wang and Schwarz 2009 a, b; Wang et al., 2011; Hsieh et al., 2016). To further test the RNAi efficiency, we also performed Western blotting on HEK cells transfected with the same RNAi, which validates the results in neurons.

Although the Reviewer is right about using CRISPR isogenic controls and multiple lines in the iPSC field, the purpose of most of those studies is to model a human disease. Because there are line-to-line variations, the best way to control for background variation is to reverse a point mutation to wild-type or to remove this pathogenic gene by CRISPR in the same patient line, or include multiple patients' lines especially when the pathogenic gene is unknown (idiopathic). However, the purpose of our study is simply to genetically knock down metaxins and traks in a healthy human neuronal line and observe their impact on mitochondrial movement. Therefore, all our RNAi studies should be done in the same wild-type line. RNAi technology remains a powerful and convenient method to knockdown genes in neuronal cultures. Many researchers still use this method to knockdown genes in iPSC-derived neurons. For examples, RNAi was used on dopaminergic neurons (Tabata et al., 2019) and motor neurons (Shelvkov et al., 2019) in these recent publications.

Reviewer #4 (Remarks to the Author):

The authors have addressed my only significant concern.

Refs:

Armstrong, L.C., Saenz, A.J., and Bornstein, P. (1999) Metaxin 1 interacts with metaxin 2, a novel related protein associated with the mammalian mitochondrial outer membrane. *Journal of cellular biochemistry* 74, 11-22.

Chung-Han Hsieh, Atossa Shaltouki, Ashley E. Gonzalez, Alexandre Betterncourt da Cruz, Lena Burbulla, Erica St. Lawrence, Birgit Schuele, Dimitri Krainc, Theo Palmer, and Xinnan Wang. (2016) Functional impairment in Miro degradation and mitophagy is a shared feature in Familial and Sporadic Parkinson's Disease. *Cell Stem Cell* 19: 709-724.

Shelvkov et al. (2019) A High-Content Screen Identifies TPP1 and Aurora B as Regulators of Axonal Mitochondrial Transport. *Cell Reports*.

Tabata, Y., *et al.* (2018) T-type Calcium Channels Determine the Vulnerability of Dopaminergic Neurons to Mitochondrial Stress in Familial Parkinson Disease. *Stem cell reports* **11**, 1171-1184.

Wang, X., and Schwarz, T.L. (2009a) The mechanism of Ca^{2+} -dependent regulation of kinesin-mediated mitochondrial motility. *Cell* **136**: 163-174.

Wang, X., and Schwarz, T.L. (2009b) Imaging axonal transport of mitochondria. *Methods in Enzymology* **457**: 319-333.

Wang, X., Winter, D., Ashrafi, G., Schlehe, J., Wong, Y., Selkoe, D., Rice, S., Steen, J., LaVoie, M., and Schwarz, T.L. (2011) PINK1 and Parkin target Miro for phosphorylation and degradation to arrest mitochondrial motility. *Cell* **147**: 893-906.

REVIEWERS' COMMENTS

Reviewer #1 (Remarks to the Author):

The revised manuscript improved and contain now important controls. Some open issues like the relationship of the adaptor complexes to SAM50 remain. Overall, the finding that Metaxins play a role as part of adaptor complex in the transport of mitochondria is very interesting for a broad readership.

Reviewer #3 (Remarks to the Author):

My concern regarding a Western blot control for iPSCs experiments was now explained in a lengthy explanation for the experimental decision. While these arguments are understandable, it needs to be noted that the standard in the field (despite rare publications using this technique) is different. The fluorescence analysis provided in ext. Fig. 6 shows very variable levels and variable cell sizes in C and high variabilities following quantifications in d and further adds to these concerns.

Response to reviewers

Comments:

Reviewer #1 (Remarks to the Author):

The revised manuscript improved and contain now important controls. Some open issues like the relationship of the adaptor complexes to SAM50 remain. Overall, the finding that Metaxins play a role as part of adaptor complex in the transport of mitochondria is very interesting for a broad readership.

Response: We agree with the reviewer that SAM50 will be an interesting molecule to study next. We thank the reviewer for recognizing the importance of this manuscript.

Comments:

Reviewer #3 (Remarks to the Author):

My concern regarding a Western blot control for iPSCs experiments was now explained in a lengthy explanation for the experimental decision. While these arguments are understandable, it needs to be noted that the standard in the field (despite rare publications using this technique) is different. The fluorescence analysis provided in ext. Fig. 6 shows very variable levels and variable cell sizes in C and high variabilities following quantifications in d and further adds to these concerns.

Response: we acknowledge that CRISPR mediated knockout and knock downs have become more and more prevalent to perform loss-of-function genetic manipulations for the iPSCs field. For these particular experiments, we needed to perform sparse transfection in order to record the movements of mitochondria and to determine the directionality of movements. Therefore, we opted to use the siRNA for these experiments. The reviewer is correct that the variable knockdown likely accounts for the high variability shown in ext. Fig. 6, which is not surprising for the siRNA knockdown.

Compared to CRISPR, RNAi is fast and convenient. It enabled us to quickly knockdown 4 genes and observe phenotypes in mitochondrial transport, which clearly demonstrate that the respective genes were knocked down in those axons. In addition, this figure complements the main finding of the paper which is based on complex genetic interaction (with combinations of multiple CRISPR KO mutants) and in vivo live imaging using the genetically tractable *C. elegans* model.